# Programmable memristors with two-dimensional nanofluidic channels

Abdulghani Ismail [1,2], Gwang-Hyeon Nam[1,2], Aziz Lokhandwala[1,2], Siddhi Vinayak Pandey[1,2], Kalluvadi Veetil Saurav [2,3], Yi You[1,2], Hiran Jyothilal[1,2], Solleti Goutham [1,2], Ravalika Sajja[1,2], Ashok Keerthi [2,3,4] & Boya Radha [1,2,4] ✉

Nanofluidic memristors, obtained by confining aqueous salt electrolyte within nanoscale channels, offer low energy consumption and the ability to mimic biological learning. Theoretically, four different types of memristors are possible, differentiated by their hysteresis loop direction. Here, we show that by varying electrolyte composition, pH, applied voltage frequency, channel material and height, all four memristor types can emerge in nanofluidic systems. We observed two hitherto unidentified memristor types in 2D nanochannels and investigated their molecular origins. A minimal mathematical model incorporating ion–ion interactions, surface charge, and channel entrance depletion successfully reproduces the observed memristive behaviors. We further investigate the impact of temperature on ionic mobility and memristors characteristics. In this work, we show that the channels display both volatile and non-volatile memory, including short-term depression akin to synapses, with signal recovery over time. These results suggest that nanofluidic devices may enable new neuromorphic architectures for pattern recognition and adaptive information processing.

Most memristors to date are solid-state devices made from metal oxides, perovskites, sulfides, organic compounds, electrode materials, or 2D materials, with their performance significantly influenced by composition, structure, and engineering[1]. Unlike traditional computing and memory, which rely on electrons and holes, natural memristors (neurons and synapses) operate with ions and neurotransmitters, offering superior energy efficiency and memory capabilities[2]. Although various attempts have been made to develop liquid-state memristors employing liquid metals[3], electrolytes[4], hydrogels[5], and ionic liquids[6–10], challenges like non-miscibility with aqueous media and broad applied potential windows have hindered progress. We have recently introduced an electrolyte-based liquid-state memristor[11] that overcomes these limitations[12]. This memristor features nanoconfined 2D slit-shaped channels with pristine MoS2 or activated carbon walls. This setup allows for the use of simple aqueous electrolyte salts and significantly reduced voltages, within the water oxidation window[11]. In

parallel, Xiong et al.[13] have developed an ionic memristor, with crossing-type memory, based on ionic interaction with confined polyelectrolyte in nanopipettes. More recently, Emmerich et al.[14,15] have developed a nanofluidic memristor based on liquid blisters formation within device interfaces.

The pursuit of multifunctional devices capable of serving diverse roles in memory systems and computing is highly desirable, as they can reduce energy consumption, simplify circuit design, and reduce the fabrication time/costs[16–18]. However, current approaches face limitations, and the activation processes for these functionalities can be irreversible or incompatible within the same material system. Sun et al.[19,20] theoretically proposed the existence of four possible memristor loop styles, but no single device or device architecture, neither in the solid state nor in the liquid state, has yet demonstrated all four loop styles. In this work, we investigate the existence of four different memristor IV-loop styles using 2D nanochannels fabricated from hBN

[1]Department of Physics and Astronomy, School of Natural Sciences, The University of Manchester, Mancheste, UK. [2]National Graphene Institute, The University of Manchester, Manchester, UK. [3]Department of Chemistry, School of Natural Sciences, The University of Manchester, Manchester, UK. [4]Photon Science Institute, The University of Manchester, Manchester, UK. ✉e-mail: radha.boya@manchester.ac.uk

and MoS$_2$. We elucidate the necessary conditions for the manifestation of each style and delve into the underlying mechanisms, along with short- and long-term memory within these nanochannels.

## Results and discussion

### Device architecture and experimental setup for ionic transport measurements

To study the distinct possible memory effect, we measure ion flows through 2D nanochannels[21,22] by applying voltage with varying frequencies (Fig. 1a and Supplementary Figs. 1 and 2). These 2D nanochannels are similar to parallelepiped voids with a fixed height along their length (Supplementary Fig. 1). The walls of the 2D nanochannels are composed of atomically smooth hBN or MoS$_2$ flakes as top and bottom walls, which are separated from each other by an array of graphene nanostrips. The lateral separation of the strips determines the width of the nanochannels (typically ~120 nm ± 10 nm) (Fig. 1b), whereas thickness of the graphene nanostrips represents the height of the channels (varied between 0.7 nm and 100 nm) (Fig. 1c). These nanochannels were transferred onto a freestanding silicon nitride membrane with a pre-fabricated microhole. The nanochannel devices were positioned within a fluidic cell, which was connected to two electrolyte-filled reservoirs. We measured various electrolytes (mono-, bi-, or trivalent chloride or sulfate electrolytes) at various concentrations varying from 0.01 M to 10 M.

### Observation of distinct memristor loop types in 2D nanochannels

Figure 1 shows a typical example of various memristor effects that could be obtained using a hBN nanochannel device (channel height, $h$-2 nm) with various salt electrolytes. Non-linear pinched IV loops, either "self-crossing" (Fig. 1d, f) or "non-self-crossing" (Fig. 1e, g) at the origin, were observed. We classify these loop styles as per the nomenclature in the ref. 23 as type I and type II, respectively. Further sub-classification comes from IV curves, whether the current or conductance increases (counter-clockwise) or decreases with applied voltages (Fig. 1 and Supplementary Fig. 3). These different forms have been observed in natural neuronal systems in plants and human brain, as well as in some artificial memristive devices, e.g., tungsten filament lamp, compact fluorescent lamp, HP memristor, etc.[24–26]. In our channel devices, we observe four possible pinched loops (two of each of types I and II)[19], in the same nanochannel device, which are associated with different memory mechanisms (Fig. 1d–g). We named the type I loops 'crossing 1' (M1) and 'crossing 2' (M3), where they showed a difference in the direction of crossing (red and blue lines in Fig. 1d, f). The conductance-voltage (GV) loops of the two 'type I' crossing loops are markedly different; crossing 1 (crossing 2) shows higher (lower) conductance at negative polarity (Supplementary Fig. 3a, c). On the other hand, the 'type II' GV curves display a wing-like or inverse wing-like structure and were named 'Saturation' (M2) and 'Wien' (M4) (Fig. 1e, g). As per the defining characteristics of a typical memristor, all four styles have a pinched loop at the origin, with the area of the loops varying with frequency (Supplementary Figs. 8–10). To understand the conditions at which these four different memristor styles are observed, we collected the data of 25 devices and made a 3D plot based on different parameters of material composition, salt type, salt concentration and height of the channel (Fig. 2). This plot allows us to identify zones where one or other memristor type exists and helps in formulating hypothesis about the mechanism behind these memristors and most importantly the necessary conditions for their occurrence. Additionally, a complementary principal component analysis (PCA) was performed to visualize trends in loop styles based on multiple parameters after reducing their dimensionality to a 2D graph (Fig. 2d). The analysis revealed clustering of memristive effects, although complete segregation was not observed. This indicates that similar characteristics are shared among memristor types, despite

differences in the original five variables before PCA: salt type, valency, concentration, channel height, and channel material. In the following sections, we will analyze the mechanisms behind the memristive effects by examining the variations in individual parameters.

### Experimental conditions of the four loop styles

In MoS$_2$ nanochannels, the Wien memristor (M4) was observed at high electrolyte concentrations (>0.1 M), while at low concentrations and/or thick channels (>10 nm) crossing 2 (M3) was observed (Fig. 2c and Supplementary Fig. 18). With hBN nanochannels, we observed two distinct "self-crossing" IV curves (Fig. 1d, f) (M3 and M1): One at low concentrations and another at high concentrations of chloride salts (represented by blue and green triangles in Fig. 2a, b). The direction of the crossing is different between these two IV curves. The observation of these two crossing effects, for same electrolyte type and device, was observed across all tested chloride salts, and predominantly in multi-valent cation salts (Fig. 2b). At high concentration of monovalent chloride salts, Wien memristor (M4) was observed (red circles in Fig. 2a). Those electrolytes with sulfate anions (−2 charge) instead of chloride anions always resulted in Wien memristor with bivalent and trivalent cations (Supplementary Fig. 24).

Upon changing the electrolyte to AlCl$_3$, a distinct Type II (non-self-crossing) memristor (M2) emerged in nanochannels, characterized by current saturation and a decrease in conductance at higher voltages (Fig. 1e, Fig. 2 black squares, and Supplementary Fig. 3b). The IV curve of M2 exhibited two distinct regions: a low-resistance state (LRS) ohmic region, where the I–V characteristics displayed a linear relationship, and a high resistance statree (HRS), referred to as the limiting current region, where the current reached a saturation point, ceasing to increase or even decreasing (Fig. 1e). In some cases, a third region where the current increases again, called over limiting region. In case where the current decreased after saturation, the memristor was described to have negative differential resistance behavior (*vide infra*) (Supplementary Fig. 26). The saturation-type (M2) memristor was exclusively observed with MnCl$_2$ and AlCl$_3$ (Supplementary Figs. 23 and 25) or at very low concentrations of other ions, where it could be mistaken for capacitive current (black squares in Fig. 2). This phenomenon was not observed with other bivalent salts such as Ca$^{2+}$ and Mg$^{2+}$ (Supplementary Fig. 22). Aluminum sulfate exhibited either Wien (M4) or crossing (M3) memristor behavior instead of saturation (Supplementary Fig. 25).

All four memristive phenomena showed a dependence on frequency, where at higher frequencies, the IV-loop area decreases, a typical characteristic of memristors[24] (Supplementary Figs. 8–10). The existence of these different styles within the same device and non-random regions on the 3D plot can be intriguing for deducing the mechanism behind these memristive effects. We extended our experiments beyond channel material, salt type, and concentration by varying pH, applying concentration gradients, and changing temperature to further probe the mechanisms behind the memristive behavior, as detailed in the following sections.

### Mechanisms of the four memristive styles: case of crossing styles – M1, M3

The crossing memristor styles reported in literature were related to various mechanisms based on different conductance states in the nanofluidic systems, including differences in adsorption/desorption kinetics, hydrophobically gated nanochannels, blister formation, etc.[11,14,27–29]. Both the type I crossing loop styles that we observed in this study were primarily differentiated by the current direction, in addition to the difference in the ratio of the positive to negative current, i.e., rectification factor (Fig. 1d, f). Our 2D nanochannels are of the same height within the length of the nanochannel (Supplementary Fig. 1); however, they have asymmetry in their channel entrance. One entrance is from the device side, and the other is from a micrometric

 

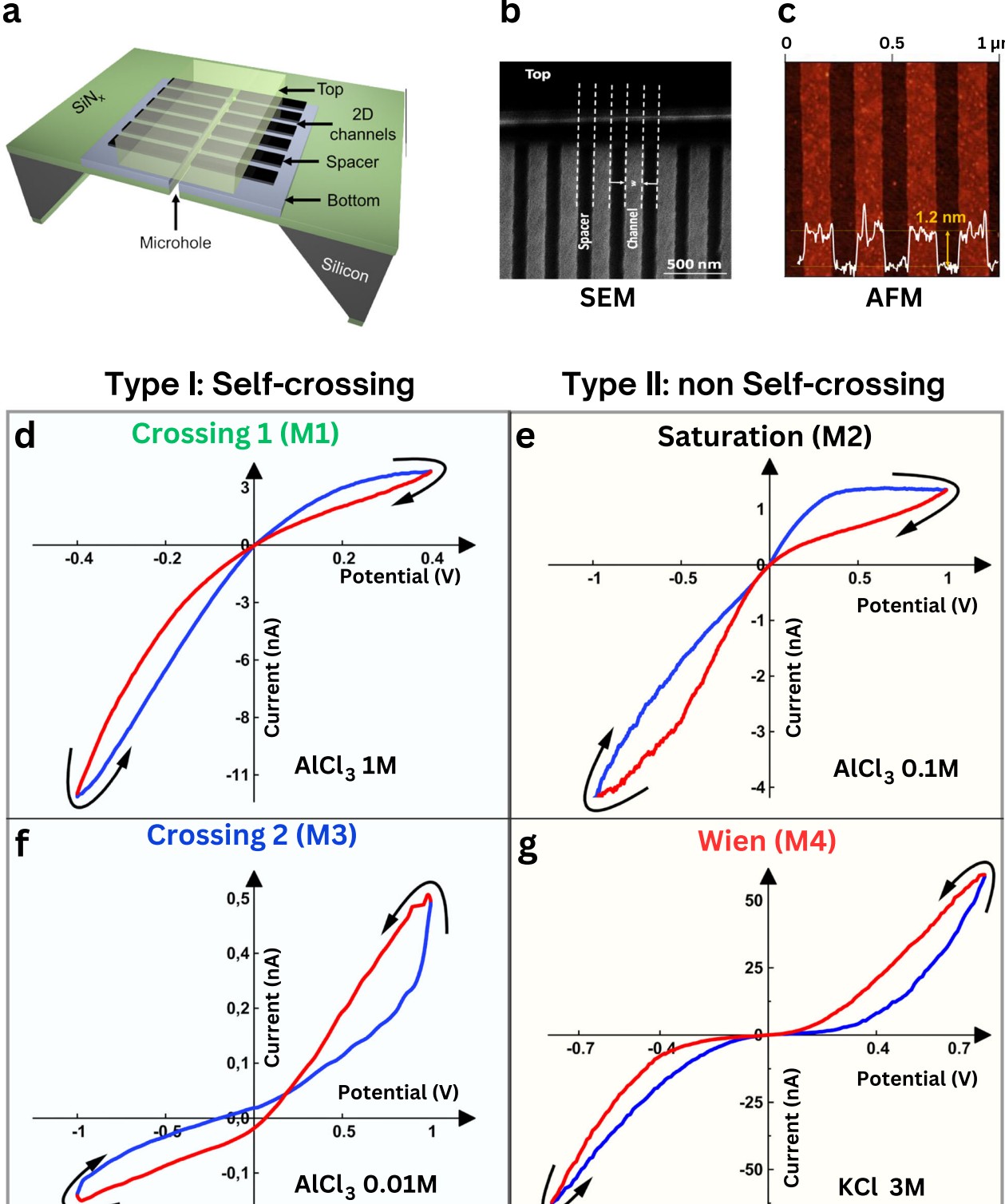

**Fig. 1 | Different memristor styles in nanochannels. a** Schematic of the device with 2D nanochannels composed of top, bottom, and spacer layers. The stack of these three layers rests over a microhole in a silicon nitride ($SiN_x$) membrane. The top and bottom layers used here are either hBN or $MoS_2$, and the spacer is graphene. An alternating voltage is applied across two Ag/AgCl electrodes in electrolyte solutions across the nanochannel. **b** Scanning electron microscopy (SEM) and **c** Atomic force microscopy (AFM) images of the spacers with a width of 130 nm and a height of 1.2 nm. **d**–**g** Electrical characterization of memristive behavior showing typical current–voltage characteristic curves observed in hBN channels, where either self-crossing (**d** and **f**) or non-self-crossing (**e** and **g**) memristor effects occur depending on the experimental conditions. The color codes of the IV curves show how they either cross (**d** and **f**) or tangentially touch (**e** and **g**) when meeting around 0 V. Current–voltage curves of (**d**) M1: crossing 1 memristor, the conductance decreases at positive polarity and increases at negative polarity. **e** M2: saturation memristor, the conductance decreases at both positive and negative high polarities. **f** M3: crossing 2 memristors, the conductance increases at positive polarity and decreases at negative polarity. **g** M4: Wien memristor, the conductance increases at both positive and negative high voltages. All the curves were obtained in a hBN device ($h = 2$ nm). Different electrolytes were used to obtain the corresponding loop styles: **d** $AlCl_3$ 1 M, **e** $AlCl_3$ 0.1 M, **f** $AlCl_3$ 0.01 M, and **g** KCl 3 M.

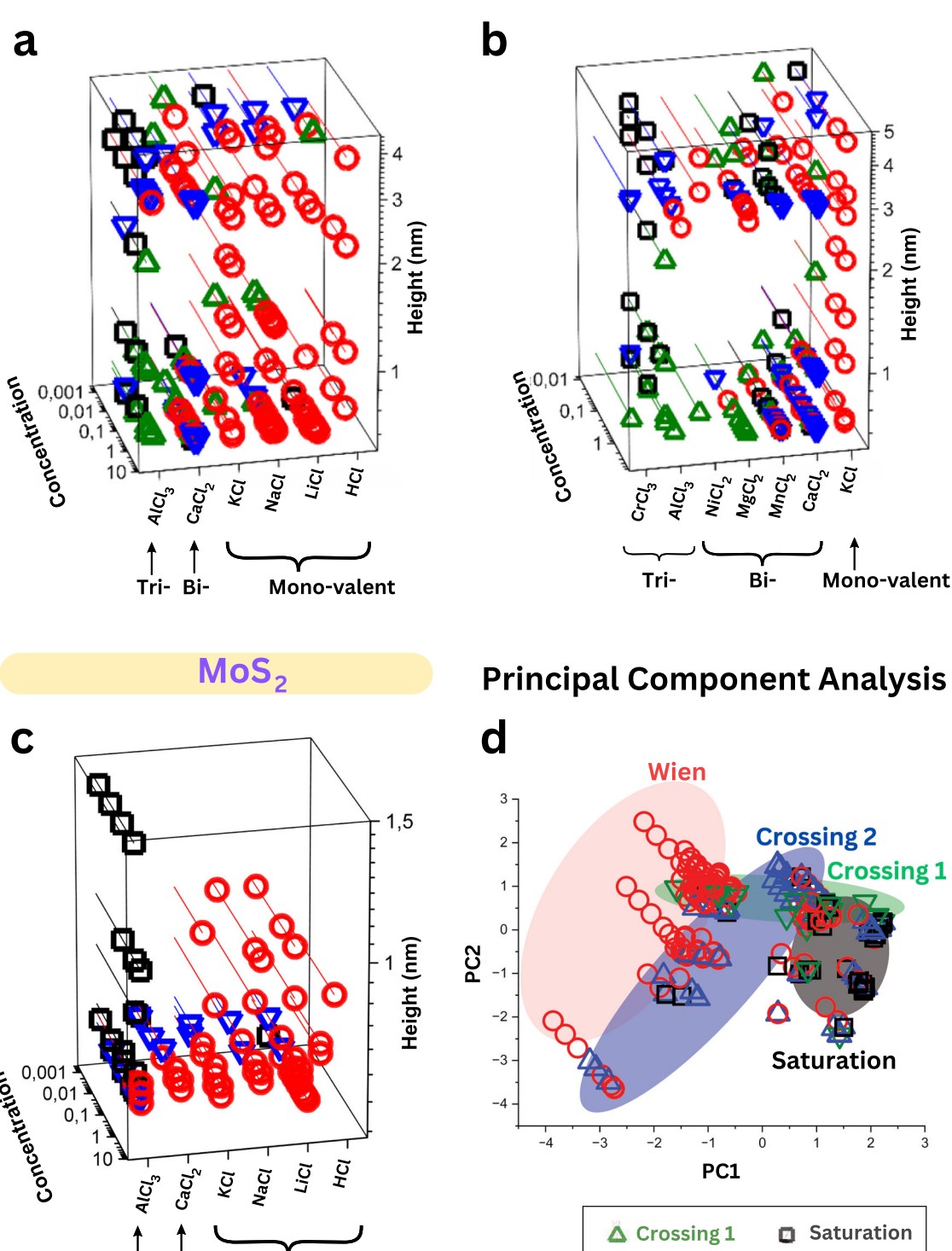

**Fig. 2 | 3D diagram and principal component analysis depicting memristive phenomena under various conditions.** The different memristor loop styles obtained from several **a, b** hBN, and **c** MoS₂ devices. The x, y, and z axes represent the salt type, salt concentration, and the nanochannel height, respectively. MoS₂ devices showed crossing 2-type (M3) memristor at low salt concentrations and for thick channels; Wien-type (M4) memristor at high concentration and thin channels; saturation-type (M2) mainly with trivalent aluminum chloride (or at very low concentrations of monovalent, where the memory is mixed with the capacitive effects). **a, b** The hBN devices showed crossing behavior at high concentration,

particularly for multivalent cations. The existence of two crossing types, one at low and the other at high concentrations, occurs in several devices, which is attributed to charge inversion (discussed in Fig. 3); this effect is not seen in MoS₂ devices. The saturation style is limited to a certain concentration range of aluminum salts and MnCl₂. **d** PCA analysis of the measured devices (hBN, MoS₂) that reduces the dimensionality from 5 dimensions (salt type, valency, concentration, channel height, and channel material) to two principal components (PC1 and PC2) to enhance the visualization of clustering patterns and underlying phenomena.

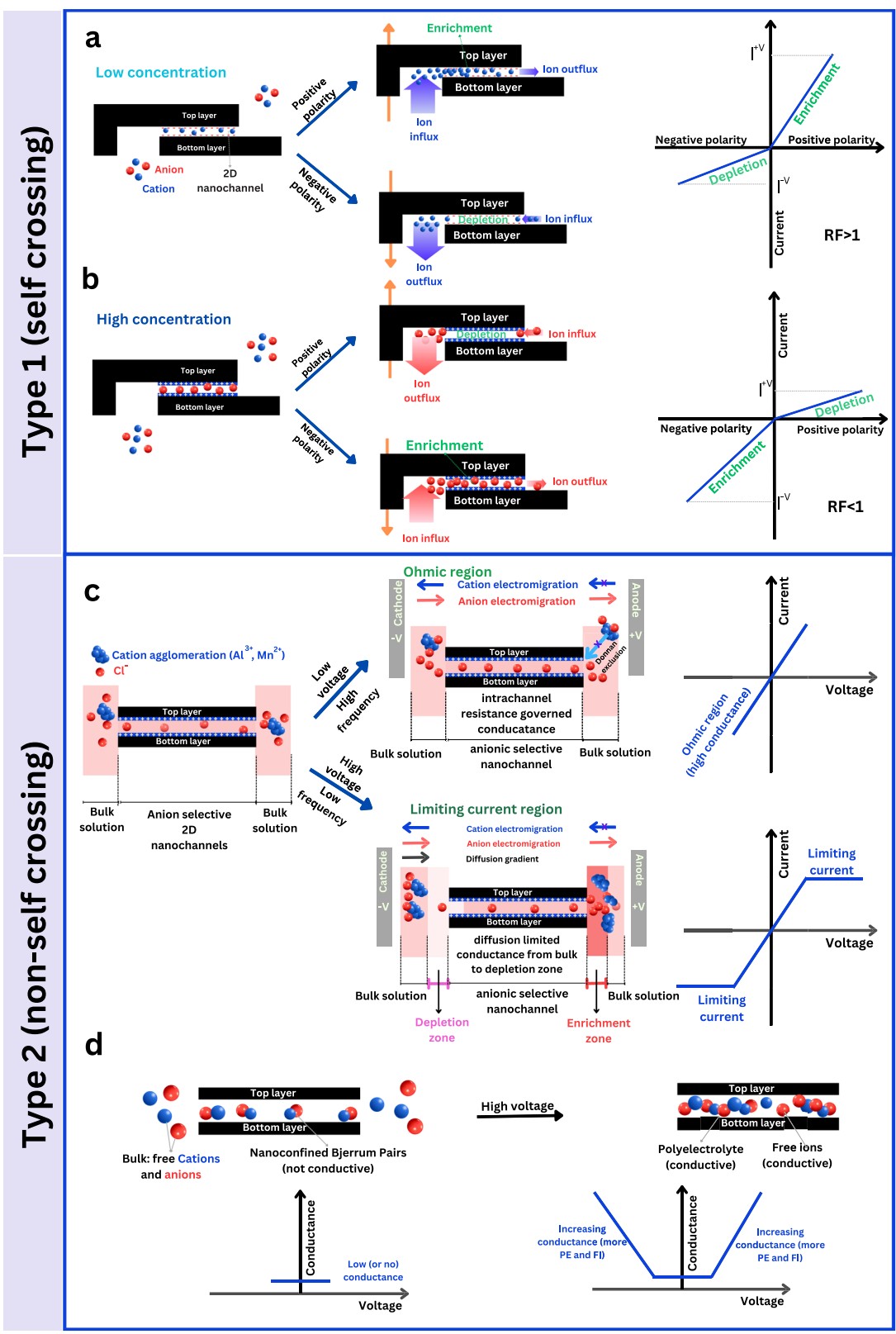

hole in the SiN$_x$/Si membrane. The difference in conductance states between positive and negative polarity (resulting in a rectification factor different than 1) is related to the channel asymmetry, where the access resistance depends on the direction of cation or anion entry to the channel (Fig. 3a, b). The variation of the rectification factor from above to below 1 (or vice versa) between different electrolyte concentrations suggests a change between high and low-conductance states upon inversion of polarities across the nanochannels. To probe the mechanism behind observing these two types of crossing, we investigated the rectification in the absence or presence of a concentration gradient to elucidate the dominant charge carrier (cation or anion)[30–32] (Supplementary Figs. 11–14).

In the case of equimolar concentration, the rectification factor differs between the two crossing loop styles as the salt concentration

**Fig. 3 | Mechanisms of the four memristive styles.** Type I (**a**, **b**) self-crossing and Type II (**c**, **d**) non-self-crossing memristive systems. **a** Crossing 2 (M3) memristor exhibits rectification (>1) at low salt concentrations due to variation of influx and outflux of cations in 2D nanochannels, resulting in enrichment or depletion depending on voltage polarity. **b** Crossing 1 (M1) memristor shows rectification (<1) at high salt concentrations, primarily with divalent and trivalent salts, leading to surface charge inversion (SCI) and anion-dominated current transport with variation of their influx and outflux due to the channel entrance asymmetry. **c** Saturation (M2) memristor loop features anion-selective nanochannels due to the SCI of the channel walls by bivalent $Mn^{2+}$ or $Al^{3+}$ ions. Initially, the ion concentrations at the nanochannel entrance are similar to the bulk, but within the channel, the anion concentration is higher, and the free cation concentration is reduced. At low

voltages or high frequencies, ion movement in the nanochannels is proportional to the applied voltage, resulting in an ohmic response. Applying high voltages and low frequencies results in variation of influx and outflux of cations and the anions across the negative surface charge of the 2D nanochannels, and thus enrichment and depletion zones at the entrance of the nanochannels. This concentration difference leads to a force opposing the electric force, and thus the resultant force to move ions (and observe conduction across the nanochannel) becomes negligible, resulting in the stabilization of the current and the decrease of conductance with further increase of voltage. This is the limiting current zone. **d** Wien (M4) memristor loop shows high resistance at low voltages due to non-conductive Bjerrum pairs and transitioning to a low-resistance state with dissociation at high voltages.

increases (Fig. 3a, b; Supplementary Fig. 16), transitioning from values greater than one (positive current higher than absolute negative one) to values less than one (or vice versa). This current asymmetry can be attributed to geometric factors or surface charge-related phenomena where the ion influx and outflux of cations (or anions) into and out of the channel strongly depend on the polarity of the applied voltage, leading to depletion or enrichment of the ions in the nanochannel[33]. Typically, as the salt concentration increases, the rectification factor tends to approach unity due to the reduced selectivity of the nanochannels[34]. However, in an intriguing departure from this conventional understanding, we observe that the rectification factor can decrease to below one with changes in salt concentration (Supplementary Fig. 16). Reports with similar observations where this phenomenon of variation of RF at high concentrations has been associated with surface charge inversion (SCI), which occurs when a charged surface or particle binds a sufficient number of counterions, resulting in a change in its net charge sign[35–37].

With a gradient of electrolyte across the nanochannels, the device showed an inversion in the rectification factor, from the value of >1 to <1 for the same gradient (△ =100), depending on the concentration range used for monovalent cation Na$^+$ (6 M: 0.06 M or 0.1 M: 0.001 M) (Supplementary Fig. 11). Using high electrolyte concentrations for monovalent salts, the negative current was more than the positive one (and vice versa). This indicates that the main charge carrier varies depending on the concentration range; it is the anion at high concentration and the cation at low concentration (see discussion in Supplementary Section 2.6). This effect was similar at high and low concentration ranges in other monovalent cations (Li$^+$), while for bivalent cations (Mn$^{2+}$ and Mg$^{2+}$), their low and high concentration ranges show only one behavior akin to monovalent ions' high concentration range (Supplementary Figs. 12 and 13). This indicated that the main charge carrier is the same at both concentrations for bivalent cations. Additional experiments confirming the role of the main charge carrier in rectification factor variation, including reversed gradients and gradient strength tuning, are presented in Supplementary Information Section 2.6 (Supplementary Figs. 12–14).

The phenomenon of SCI is illustrated in Fig. 3. SCI occurs when a multivalent ion is attracted to the channel's negatively charged surface, resulting in overscreening and a reversal of surface potential from negative to positive. This arises from ions' transverse correlation with the wall's surface charges and/or lateral correlation with each other (as described by strongly correlated liquid theory) at high salt concentrations[38]. This leads to dependence of SCI upon several factors, such as the discrete nature of charged surface sites, counter-ion valency, counter-ion to co-ion size ratio, surface charge density, nanochannel diameter, solvent dielectric constant, and pH[39–43]. Our experiments reveal SCI with bivalent ions and trivalent ions, as well as monovalent ions, when coupled with chloride co-ions (Supplementary Fig. 24). This emphasizes the significant role of ion–ion interactions in SCI. This observation of SCI with monovalent salts has not been previously reported in synthetic nanochannels[32,39,44], though it was predicted by simulation[43]. The unique 2D confinement of our

nanochannels enables this SCI to occur with monovalent cations, resulting in a hysteresis memory effect. This could also explain previous observations of SCI in biological nanochannels at high monovalent salt concentrations[45]. Furthermore, beyond cation valency and co-ion type, other parameters, such as the choice of monovalent cations[46], contribute to the nuanced nature of SCI. Notably, lithium chloride (LiCl) and sodium chloride (NaCl) exhibit a higher degree of rectification factor inversion compared to potassium chloride (KCl) (Supplementary Fig. 16). This was revealed by symmetric and gradient comparison of the RF inversion (Supplementary Figs. 12 and 16). Detailed mechanistic insights into the role of distinct conductance states, kinetic adsorption/desorption constants asymmetry, and SCI in the observation of M1 and M3 are provided in Supplementary Information Sections 1.3 and 2.7–2.10 (Supplementary Figs. 3, 8–10, 15).

Two conditions are needed for memory effects: (1) two distinct conductance states (e.g., enrichment and depletion, Fig. 3), and (2) kinetic variation causing the hysteresis pinched loop. In crossing memristors, this kinetic variation arises from ion adsorption/desorption on nanochannel walls. Prior work[11] linked cation adsorption in negatively charged activated carbon channels to long-term memory. We observed crossing 2 in hBN and MoS$_2$ channels, but lowering pH in 10 mM KCl eliminated it (Fig. S15), highlighting the surface charge's role. We propose that similar adsorption/desorption occurs after SCI, involving anions, explaining long-term memory in hBN (Figs. S8–S10). At low ion concentrations, negative surface charge leads to cation-driven conduction and crossing 2 behavior due to channel resistance and adsorption kinetics (Fig. 3a). At higher concentrations, crossing 1 arises from cation adsorption and Donnan exclusion of additional cations, allowing only anions in. The hysteresis stems from chloride adsorption-desorption dynamics and asymmetric channel entry resistance (Fig. 3b).

## Mechanisms of the four memristive styles: case of saturation loop style

The M2 loop style is a type II non-self-crossing memristor showing unipolar characteristics (Fig. 1e and Supplementary Fig. 3b). At lower voltages, the device exhibits an ohmic regime with a high conductance state, while at higher voltages, it shows a lower conductance state (Supplementary Fig. 3b). In the Ohmic regime, the conductance is governed by the intrachannel resistance. The lower the frequency of the applied voltage, the more pronounced the saturation effect becomes. The phenomenon of current saturation resembles ion concentration polarization, where external polarization by applying higher voltages for extended durations leads to an imbalance between the anion and cation flux across the selective nanochannel. This imbalance leads to the formation of depletion and enrichment concentration zones of the electrolyte at the entry and exit of the nanochannels (Fig. 3c). In this regime, the conductance is governed by the diffusion of ions to the depletion zone rather than by the presence of ions in the nanochannels[47,48]. This results in the limitation of ion movement in nanochannels, stabilizing the current, and consequently decreasing the conductance beyond a threshold voltage[49,50] (Fig. 3c).

Application of higher voltages for longer duration show the sweep between the crossing mechanism (intrachannel depletion/enrichment) and the saturation (channel entry depletion layer) (Supplementary Fig. 27). The formation of extended polarized layer, described by Rubinstein and Shtilman[51], localizes the depletion layer in the vicinity of cation exchange membrane entry rather than being extended toward the electrode and thus the sustainability of observing a constant limiting current in our nanochannel could be explained. High frequencies result in an ohmic response (no hysteresis) without a decrease in conductance, as there is insufficient time for the formation of a depletion zone and subsequent ion concentration polarization regime. This time dependence and the difference in the kinetics of formation and disappearance of the depletion and enrichment region are at the origin of the memristive behavior.

Let us now examine the conditions necessary for the observation of this phenomenon. The existence of ion selectivity across the nanochannel and a depletion regime at the entrance of the nanochannels are the two necessary conditions to achieve ion concentration polarization. As a trivalent ion, $Al^{3+}$ ion adsorption is higher on the walls of the 2D channel than bivalent ions, resulting in a higher extent of SCI and anion-selective channels. Due to this, we observe an ion concentration polarization regime rather than crossing 1 memristor. This raises the question: what factor favors the emergence of the M2 saturation IV with $AlCl_3$ and $MnCl_2$? (Supplementary Figs. 23 and 25). One explanation is that $Al^{3+}$ ions can form oligomeric and polymeric aluminum complexes in bulk $AlCl_3$ solutions[52]. Similarly, $MnCl_2$ has been shown to exist in complex dihalogenoaqueous forms rather than as individual dispersed ions in bulk water[53]. This effect is particularly pronounced in chloride salts compared to sulfate salts, as the sulfate anion has been shown to hinder the agglomeration of polymeric aluminum complexes (Supplementary Fig. 25)[54]. These oligomeric forms have dimensions larger than our nanochannels, making their entry sterically not favored, and thus can contribute to the formation of the depletion zone. Additionally, these ion agglomerates enhance the ion concentration polarization regime by strengthening the depletion layer due to their reduced mobility, steric effects, and amplification of electric field effects. These result in the observation of the non-linearity of the current at lower voltages (by up to one order of magnitude) and at higher molar concentration (more than two orders higher) than most literature reports on ICP regimes[55,56]. Notably, at these higher concentrations, the counter-ion concentration in the nanochannel is not the limiting step, and it might be more similar to the bulk concentration. Although current non-linearities and hysteresis behavior were reported in the previously cited reports[55,56], however, in this report, we show the pinched hysteresis loop with its area varying with frequency, characteristic of the distinct memory behavior. The endurance of saturation-type memristor was reflected by repetitive IVs with similar on/off ratio even after more than 250 cycles (Supplementary Fig. 28).

### Mechanisms of the four memristive styles: crossing 2 and Wien loop styles

These two styles, named M3 and M4 here, have been described in our previous report[11] with their mechanisms relating to surface adsorption/desorption process and Wien-like mechanism due to ion pairing (Fig. 3a, d). The interplay between surface charge, electrolyte composition, and anion effects in toggling between M3 and M4 styles is detailed in Supplementary Section 2.7, including transitions driven by pH and salt composition (Supplementary Fig. 15). The M3 curves typically observed at low ion concentrations, have ionic rectification which can be explained by the variation of the influx and outflux of counterions (cations in this case) due to ion selectivity inside confinement as well as due to the asymmetric entry/exit for ions (Fig. 3a). The explanation of Wien mechanism is challenging; it was first simulated by Robin et al.[57] where Bjerrum pairs were predicted to be

observed in the absence of any applied voltage, that transform to polyelectrolyte under electric field in 2D confined nanochannels. Following this, the Wien-type memory was shown experimentally in $MoS_2$ pristine 2D nanochannels, and the IV characteristics of the experiment matched the simulation, albeit with exceptionally long timescales[11]. We explained this long-term memory (minutes to hours) by taking into account the slow effective transport due to the pairing/unpairing mechanism of 2D confined ions (and by the 'stop and go' mechanism of adsorption/desorption in case of M3 loop in activated carbon nanochannels). While the direct experimental identification of Bjerrum pairs and their transformation into polyelectrolytes remains technically challenging (e.g., via spectroscopy or electron microscopy), the alignment of simulations with our experimental electrical data supports this interpretation[58].

Hysteresis in IV curves induced by the Wien effect and controlled by ion concentration propagation was reported previously[59,60]. In this case, the hysteresis occurs due to ion depletion at the bipolar membrane junction, leading to extremely high electric fields ($\sim 10^7$ V.cm$^{-1}$) which emerge after salt depletion under modest voltage bias[59,60]. In contrast, our system operates under electric fields $\sim 10^3$ V.cm$^{-1}$, and within the stable water activity range. Additionally, control experiments using pure water (without added salt) resulted in a pure capacitive IV curve without any memory or hydrolysis effects. This confirms our interpretation that the reason behind the current Wien effect under high confinement is polyelectrolyte Wien effect driven by ion dissociation rather than water splitting. Other concurrent mechanisms, such as ionic Coulomb blockade[61], surface charge ionic blockade[62], or hydrophobic gating[27], can likely be ruled out as our experimental conditions differ significantly from those under which these effects have been observed. The nanoscale confinement and the simultaneous potential application during characterization in the liquid state pose technical challenges for experimental identification of Bjerrum pairs/polyelectrolyte transformations. Additionally, we demonstrated that M3 and M4 effects occur in both hBN and $MoS_2$ devices, and can be switched in the same channel by adjusting salt concentration and pH (Fig. S15).

### Impact of nanochannel height on memristive behavior and confinement-dependent phenomena

The height of the nanochannel is fundamental to achieving 2D confinement[63] and has a significant effect on all the memory styles and their existence. All thick nanochannels (>10 nm) exhibited very small pinched loop tending to an ohmic behavior (Supplementary Figs. 18–19, 21). For Type I self-crossing memristors (M1 and M3), this phenomenon can be attributed to the absence of intrachannel resistance limitation, which is essential for observing different conductance states and current rectification. In the case of Wien memristors, while simulations[57] suggested that Bjerrum pairs, and consequently M4 memristors, should only exist under confinement conditions of up to 2 nm, the experimental observations extended their presence to channels as wide as 10 nm. This discrepancy may be due to the formation of Bjerrum pairs in the water layers adjacent to the channel walls. The reduced dielectric constant in these water layers could facilitate this pairing near the walls[60,64,65].

The M2 saturation memristor showed non non-negligible loop area for thick channels up to 20 nm (Supplementary Figs. 9 and 21). This behavior relates to its underlying mechanism, which involves external polarization. Specifically, the depletion zone at the nanochannel entry persists, particularly at low ionic concentrations (C < 100 mM). At higher concentrations (C > 1 M), the pinched hysteresis diminishes and eventually disappears.

The effect of nanochannel height is also evident in the behavior of the crossing 1 memristor, which relies on SCI. This memristor is primarily observed in nanochannels with heights below 2 nm at high concentrations of $Al^{3+}$. Notably, the effect becomes more pronounced

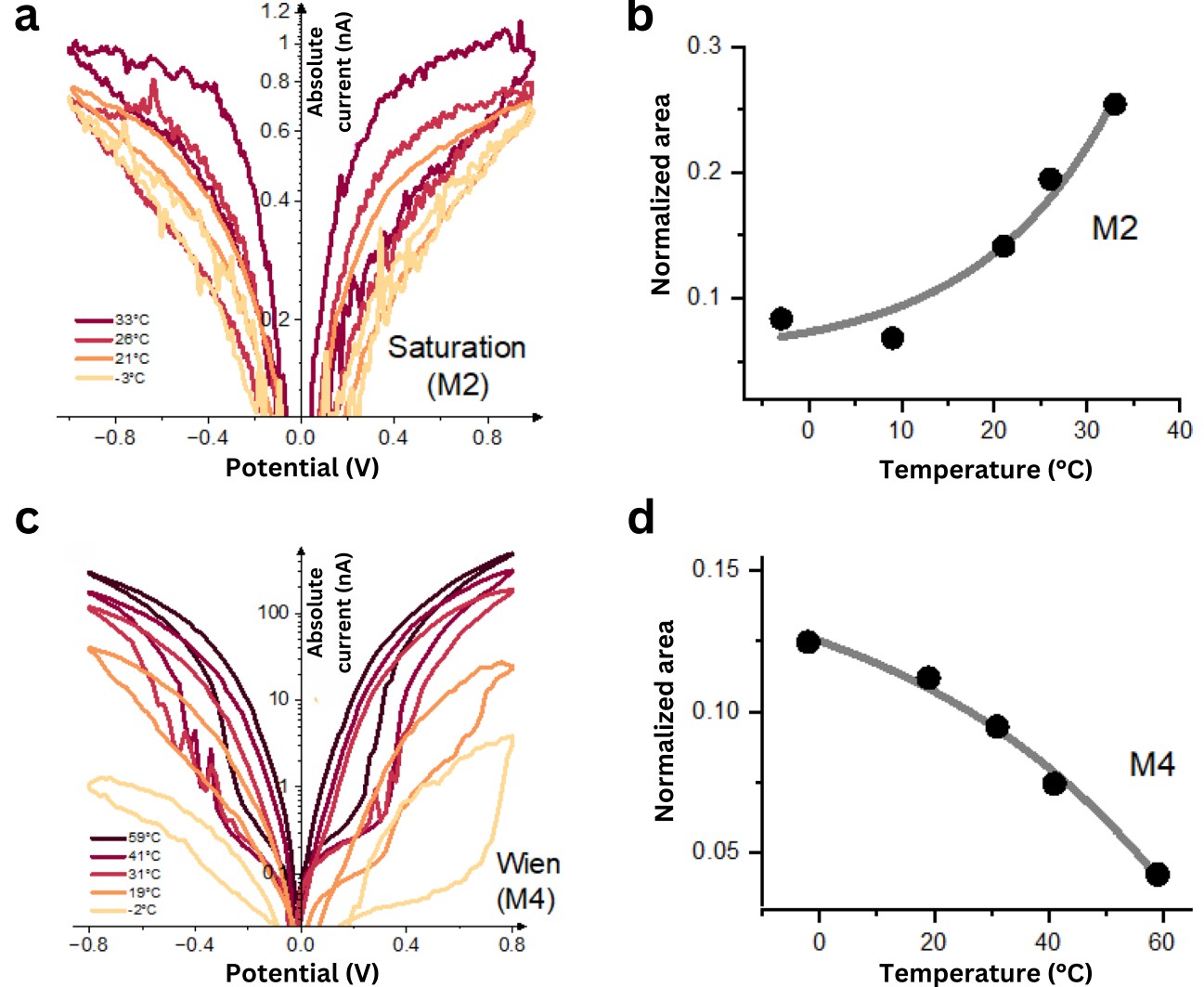

**Fig. 4 | Temperature effect on type II non-self-crossing memristors.**
**a** Current–voltage curves showcasing the Wien-type memory (M4) with a hBN device ($h = 2\,nm$, HCl 1 M). **c** Current–voltage curves showcasing the Saturation-type (M2) memristor with a hBN device ($h = 1\,nm$, AlCl$_3$ 0.5 M). Normalized area variation vs. temperature for **b** saturation type (M2) and **d** Wien-type (M4) memristors display contrasting effects with temperature. Normalization is done by dividing the loop area by the highest value of absolute conductance ($|I|_{max}/V_{max}$).

at lower concentrations in highly confined nanochannels. The relationship between confinement and the concentration required to observe SCI has been discussed in the literature[32,66], with SCI occurring at low concentrations in highly confined nanochannels.

**Temperature dependence of non-crossing memristor dynamics**
To delve deeper into the mechanisms underlying the observed memristive phenomena, we analyzed the impact of temperature variation on the IV curves of non-self-crossing type II memristors (Fig. 4). Temperature fluctuations can influence memristive effects by modifying factors such as ion mobility, solvation, hydration, Debye length, and ion–surface and ion–ion interactions. In the case of the Wien memristor (M4), increasing the temperature resulted in reducing the hysteresis, eventually eliminating it at high temperatures and thus decreasing the system's memory (Fig. 4c, d), while for the saturation memristor (M2), the opposite behavior was observed where higher temperatures intensified the hysteresis area and thus memory effect (Fig. 4a, b).

We can explain observations of temperature variation by correlation with the mechanism behind both memristor styles. In the Wien-type memristor (M4), Robin et al.[57] theoretically correlated the formation of Bjerrum pairs with the reduced temperature, noting that lower temperatures promote their formation. Our experimental results confirmed this hypothesis, where a larger disparity between the low and high conductive states and a stronger memristive effect were observed by decreasing the temperature. As temperature rises, the formation of Bjerrum pairs is not favorable as thermal fluctuations become more favorable than the electrostatic interactions, increasing the presence of free ions and diminishing the presence of Bjerrum pairs and polyelectrolyte (Supplementary Fig. 34b). On the other hand, in the saturation-type memristor (M2), temperature significantly influences ionic mobility, with higher temperatures increasing diffusion rates and expediting the formation of the depletion zone at the channel's entry, thereby enhancing loop formation (Supplementary Fig. 34a).

**Summary of the mechanistic determinants of memristive behavior**
To summarize the various effects that determine the memory type: M4 (Wien effect) memristor is observed when the ions interact with themselves more than with the surface charge, allowing the applied electric field to dictate ion dynamics, particularly in high monovalent electrolyte concentrations where the channel behaves as a near-

neutral system. This fact is consolidated by the shifting of the governing effect from crossing 2 (M3) to Wien (M4) when the surface charge is reduced by decreasing pH at low electrolyte concentrations (Supplementary Fig. 15). When surface charge effects dominate, the occurrence of M1, M2, or M3 depends on the surface-ion interaction intensity:

1. Monovalent cations exhibit weaker interactions than multivalent cations with the nanochannel walls, leading to predominant M4 behavior at high concentrations. However, at lower concentrations, surface effects can become more significant, leading to crossing-type behavior (M1 or M3).

2. Multivalent cations interact more strongly than monovalent cations with the surface charges, leading to concentration-dependent transitions among M1, M2, and M3. In particular:
   a. M3 (Crossing 2) transitions to M1 (Crossing 1) due to SCI, where the main charge carrier (anion or cation) switches. The observation of rectification due to asymmetric conditions and kinetics governed by variable adsorption/desorption dynamics is at the basis of their occurrence.
   b. M2 (Saturation) is observed in cases where ion concentration polarization plays a dominant role, particularly with multivalent cations that have polymeric forms ($Al^{3+}$ or $Mn^{2+}$), which promote SCI and depletion-layer effects at the channel entrance at high voltages.

All four effects necessitate extreme confinement; otherwise, the system will not show any memory. This systematic interplay between confinement, electrolyte concentration, and surface charge effects highlights the tunability of the nanofluidic memristor, and controlled variations in pH, electrolyte type, and confinement conditions can enable transitions between different memristor behaviors. It should be noted that the coexistence of several phenomena (M1–M4) at the same time, depending on the experimental conditions, complicates the interpretation of the observations. One effect might mask another, and the situation is a balance between the effects. For example, SCI is clearly manifested in hBN devices; however, in $MoS_2$, SCI is masked by the strong Wien effect and could be seen only from the variation of the rectification factor.

## Minimal theoretical model capturing multiple memristive loop types

Building on the qualitative insights described above, we propose a minimal theoretical model that quantitatively captures the memristive effects for the four observed loop types (M1, M2, M3, and M4), as detailed in the Supplementary Information. This model is an extension of the minimal model previously developed by Robin et al.[11], now including additional parameters explicitly accounting for ion–ion and ion–surface interactions, and reservoir depletion effects. By tuning these parameters within the same framework, we successfully simulated all four experimentally observed memristive loops that coexist in the same device but to different extents (Fig. 5). This unified theoretical approach helps in the quantitative explanation of the diverse memristive behaviors observed.

From the preceding discussions, the memory in 2D nano/angstrom channels is governed by ion–ion interactions, ion-channel wall interactions, and channels' entrance depletion. Each of these interactions has a competing nature and takes place at varying but comparable timescales. Adsorption ($\alpha_A$, $\beta_B$) and desorption ($l_A$, $l_B$) rates of anions and cations to the wall charges regulate the ion-channel wall interactions (Fig. 5). The interaction strength parameter, $\delta_0[f]$, governs the ion–ion interactions, where the Wien effects cause a voltage dependence (Fig. 5a, e). The external coupling $\gamma$ governs the depletion of ions occurring exterior to the channel, explaining the saturation-type memristor (M2) (Fig. 5a, c). Further discussions about each term are in the Supplementary Section 1.

## Synaptic plasticity and memory control in nanofluidic memristors

Biological synapses and neurons are natural counterparts to nanofluidic memristors, facilitating memory processing in the brain. We attempt to emulate biological processes, especially at the level of synaptic conduction, using our 2D nanochannels, such that an increase in nanochannel conductance resembles an increase in the synaptic weight. Various memristive behaviors observed in our artificial 2D nanochannels draw parallels with synaptic phenomena observed in biological neural systems. We report volatile (short-term) and non-volatile (long-term) memories (Fig. 6) as well as potentiation and depression behaviors.

Short-term potentiation (STP) is a form of volatile memory that is characterized by an initial increase in synaptic conductance upon stimulation, followed by a spontaneous decay once the stimulus is removed. We observed two cases of STP:

Case 1 (Fig. 6a): Low read, high write voltage. This behavior is reminiscent of excitatory postsynaptic currents (EPSCs) in biological neurons, which reflect synaptic activation and contribute to temporal signal processing and synaptic filtering. The volatile nature of this behavior is inferred from the return to the initial conductance level after relaxation at zero volts (Fig. 6a). This response is observed in M4, where the underlying mechanism involves the formation of a polyelectrolyte upon the application of a writing stimulus, which subsequently dissipates when the device is relaxed at zero volts.

Case 2 (Fig. 6b): High read voltage, relaxation at zero volts. This type of STP is observed in the M2 memristor, where a high reading voltage also acts as the writing voltage. After relaxing the device at zero volts, reapplying the high voltage briefly increases conductance before it returns to baseline. This is likely due to temporary disruption and recovery of the depletion zone at the channel entrance, which reforms once the voltage is reapplied.

## Short-term depression (STD) involves an initial decrease in conductance upon stimulation, followed by recovery

Case 1 (Fig. 6c): Low read, high write voltage. This behavior, seen in M1, resembles inhibitory postsynaptic currents (IPSCs) that modulate neural activity and prevent overstimulation. It likely stems from ion depletion and desorption under positive bias, reducing conductance. While most memory fades within minutes, a minor long-term effect remains, as in crossing 2 (Fig. 6g, h). However, in this case, the long-term effect is negligible, likely due to high electrolyte concentration enabling rapid ion diffusion that erases lasting changes.

Case 2 (Fig. 6d): High read voltage relaxation at zero volt. The results are opposite to Fig. 6b. The inhibitory response in the Wien memristor mimics $Na^+$ current decay after axonal depolarization[67]. Current inhibition scales with relaxation time, indicating exponential decay from gradual polyelectrolyte dissociation (inset of Fig. 6d).

Long-term potentiation (LTP), a cornerstone of learning and memory in biological systems, is effectively represented in Fig. 6e and f through non-volatile memory behaviors. These curves demonstrate sustained increases in conductance persisting from minutes (Fig. 6e, M3) to days (Fig. 6f, M3, 74 h). Here, applying an erasing negative polarity voltage resulted in the loss of stored memory and a return to the low-conductance state (Fig. 6f). This prolonged retention of elevated conductance levels even when the device is disconnected, closely aligns with biological synaptic strengthening, which underlies memory formation and long-term learning. These stable states highlight the potential of our 2D nanochannels to emulate synaptic plasticity and enable in-memory information storage. Figure 6g shows both short-term and long-term potentiation in the M3 memristor, governed by writing pulse duration. Short pulses produce transient conductance (short-term memory), while longer pulses induce a short-term part that disappears after a short duration in the range of tenth of seconds, as well as lasting conductance changes (long-term memory).

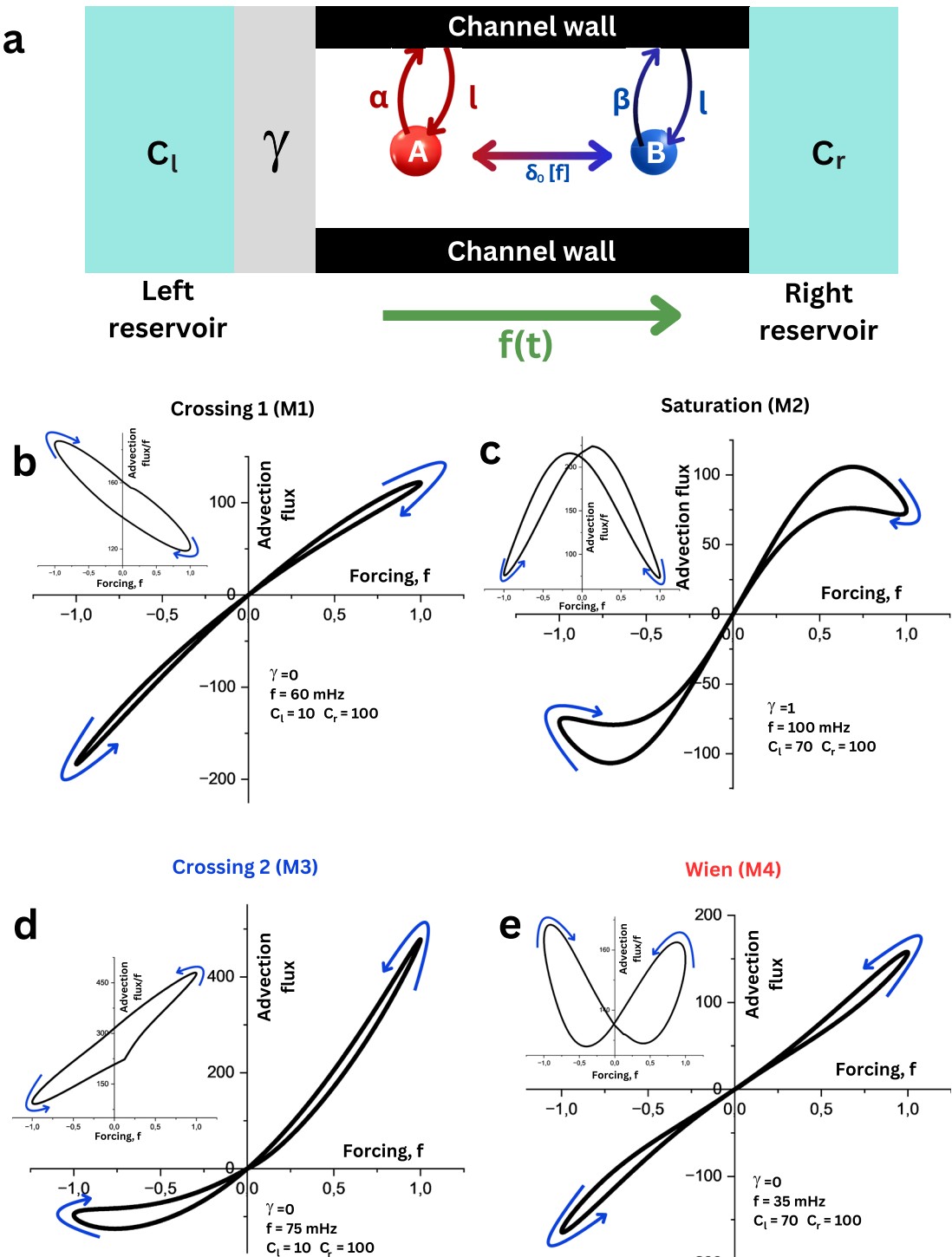

**Fig. 5 | Simulation of four memristive loops using a minimal model. a** Schematic illustrating the various parameters used in the differential equations governing ion interactions inside the nanofluidic channel (See Supplementary Section 1). These include the adsorption ($\alpha_A$, $\beta_B$) and desorption ($l_A$, $l_B$) rates of anions and cations to the wall charges, the ion–ion interaction term ($\delta_0[f]$), and channel entrance depletion parameter ($\gamma$). Under an external forcing function $f(t)$, cations and anions migrate in opposite directions from two reservoirs with different concentrations ($c_r$ and $c_l$), reflecting the asymmetric entrance of the nanochannel. **b**–**e** and their insets show the corresponding current–voltage and conductance-voltage curves, respectively, obtained using the equations in the Supplementary Section 1 by varying the interaction parameters. This results in four distinct memory effects (M1–M4). The specific parameters used for each plot are indicated within the figure panels.

This dual behavior is essential for neuromorphic systems combining working memory and stable learning.

The device shows distinct memory behaviors under the same voltage depending on the electrolyte, highlighting the role of ion dynamics and surface interactions. Long-term memory (M3) results from charge adsorption on nanochannel walls and can be erased by

applying reverse polarity. Short-term memory (M2, M4), driven by ion depletion or polyelectrolyte formation, relaxes quickly once the voltage is off. This contrast between surface charge- and ion dynamics-based mechanisms enables control over memory duration. In Wien memristors, memory resets via zero-voltage relaxation, while in crossing memristors, it requires opposite polarity (Supplementary

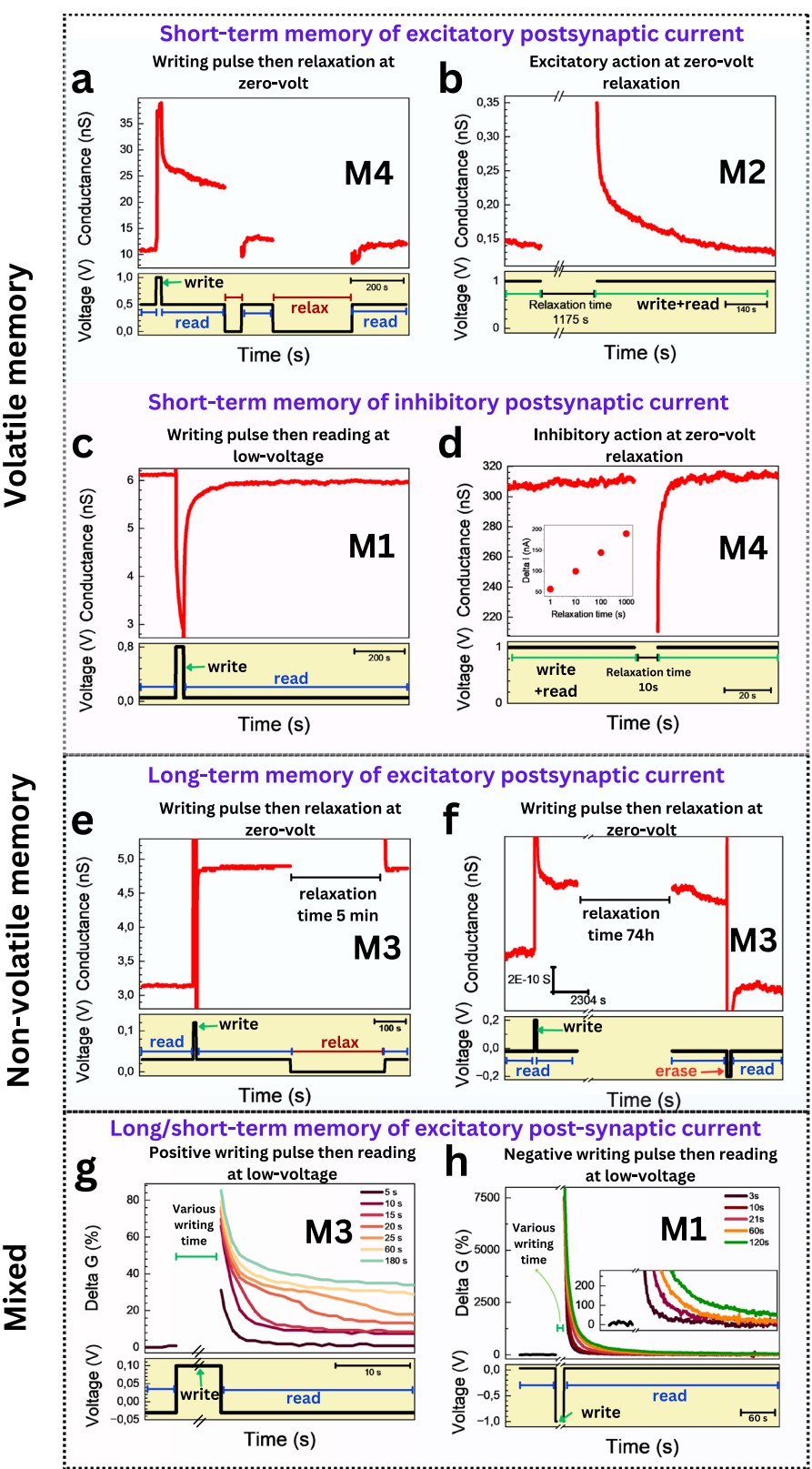

Figs. 30–33, Fig. 6g). Both memristors response depends on the number of pulses, duration, and voltage history, offering tunable functions for neuromorphic computing.

This dynamic behavior closely mirrors biological synapses (Fig. 7a), where the excitability of the postsynaptic membrane is influenced by presynaptic activity. Such short-term synaptic plasticity typically occurs over milliseconds to minutes and is classified as depression or facilitation depending on whether the second post-synaptic potential is smaller or larger than the first[68]. In our artificial nanochannels, we mimicked presynaptic excitation by applying con-tinuous current pulses (200 nA, 30 s), followed by a relaxation period and a second excitation. We observed that the amplitude of the second current response decreased relative to the first, indicating short-term depression modulated by relaxation time (Fig. 7b, c). The conductance

**Fig. 6 | Volatile and non-volatile memory. a** Excitatory action in a MoS$_2$ device ($h$ = 0.7 nm, 1 M KCl) demonstrating a Wien-type memory (M4). The retained memory after applying a positive potential pulse of 1 V is partially erased and then completely erased after relaxation at 0 V for 1 min and 5 min, respectively, transitioning the system from low-resistance state (LRS) to high resistance state (HRS). The reading voltage is 500 mV. **b** Excitatory action in a hBN device ($h$ = 0.7 nm, AlCl$_3$ 10 mM) demonstrating saturation-type memory (M2) after a 20-min relaxation at 0 V, read at 1 V. **c** Inhibitory action in a hBN device ($h$ = 2 nm, AlCl$_3$ 1 M) demonstrating a crossing 1 type memory (M1) after a writing pulse of 0.8 V, read at 50 mV. Residual long-term memory persists, as the conductance does not return to its original value. **d** Inhibitory action in a hBN device ($h$ = 1.4 nm, 3 M KCl) showing Wien-type memory (M4). The device is relaxed at 0 V and read at 1 V. The current magnitude variation relative to its equilibrium value is plotted against relaxation time in semi-log coordinates (inset). This case is the opposite of (**b**). **e** Non-volatile

memory in MoS$_2$ device ($h$ = 0.7 nm, 0.01 M KCl), showing a crossing 2 type memory (M3). The retained memory after a positive pulse of 0.12 V is not erased after relaxation at 0 V for 5 min. The reading voltage is 30 mV. **f** Non-volatile memory in a MoS$_2$ device ($h$ = 0.7 nm, 0.1 M KCl), showing a crossing 2 type memory (M3). The retained memory after a positive pulse of 0.2 V is not erased after relaxation at 0 V for ~3 days. The reading and erasing voltages are −20 mV and −200 mV, respectively. **g** Coexistence of short- and long-term memory in a MoS$_2$ device ($h$ = 0.7 nm, 0.01 M KCl), showing a crossing 2 type memory (M3). Varying writing pulse durations lead to both short (<20 s) and long-term memories. Reading is done at −0.03 V, whereas writing is done with +0.1 V pulses. **h** Coexistence of short- and long-term memory in a hBN device ($h$ = 0.7 nm, CaCl$_2$ 0.3 M), showing a crossing 1 type memory (M1). Varying writing pulse duration leads to both short (<60 s) and long-term memories; however, the short-term memory was predominant compared to (**g**). Read at 0.03 V, written at −1 V.

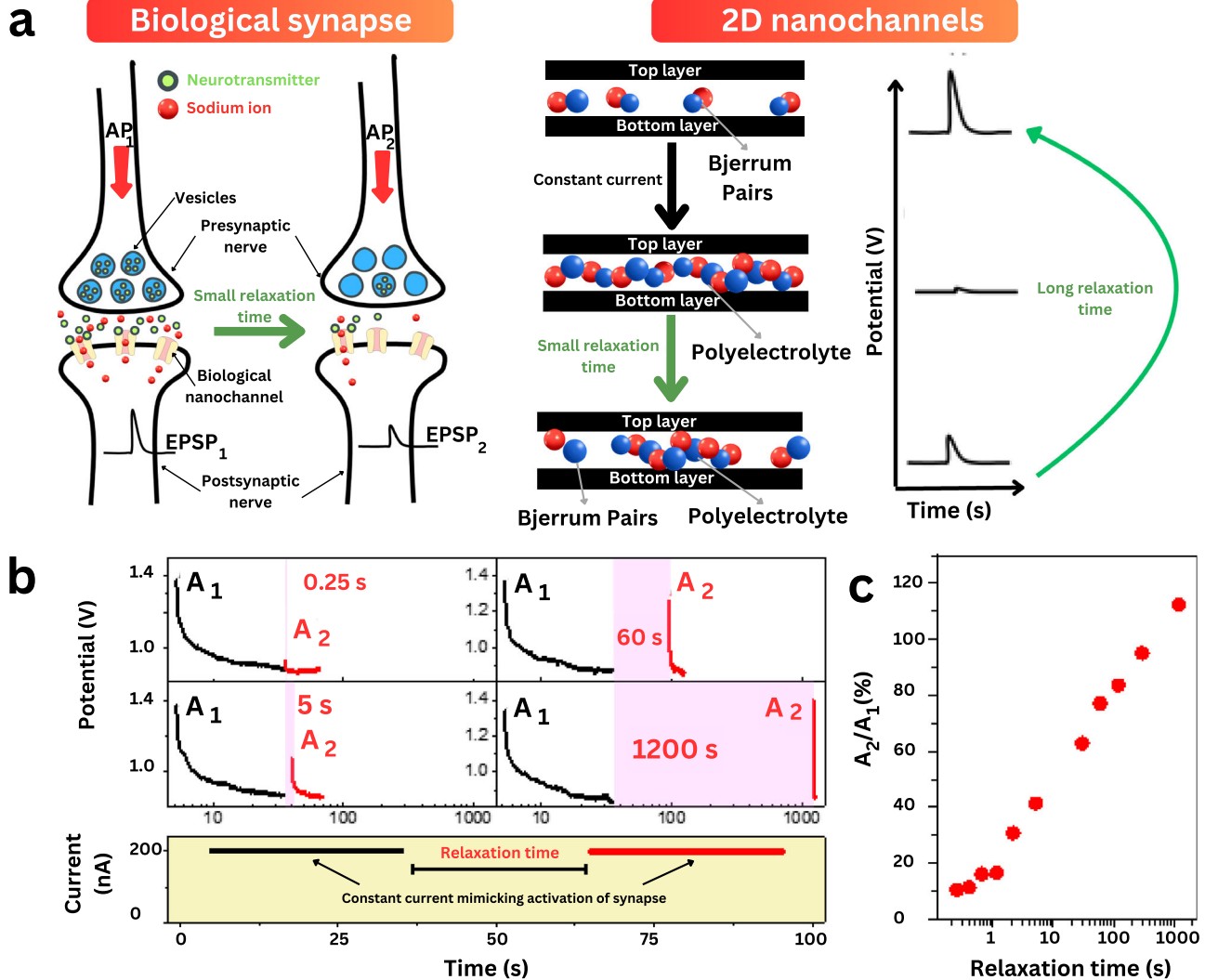

**Fig. 7 | Nanochannels mimic biological ion channels and synapses. a** Schematic of the short-term depression in biological synapses and artificial nanochannels. The left panel shows two consecutive action potentials (AP$_1$ and AP$_2$) triggering neurotransmitter release from vesicles within the presynaptic nerve terminal, subsequently generating excitatory postsynaptic potentials (EPSPs) in the postsynaptic neuron. The amplitude of the second EPSP is affected by the relaxation time between the subsequent action potentials. The right panel shows the mechanism of the short-term depression in 2D nanochannels, where the potential difference obtained after application of subsequent constant currents is related to the

relaxation time between the currents. When the current is applied, the polyelectrolyte is formed, and its dissociation to Bjerrum pairs, similar to the initial conditions, necessitates a sufficiently long relaxation time. **b** EPSP variation in nanochannels under a constant current of 200 nA for 30 s separated by different relaxation times (0.25 s, 5 s, 60 s, and 1200 s), simulating short-term depression observed in postsynaptic neurons. **c** Quantification of short-term depression by the ratio of the second EPSP peak (A2) to the first EPSP peak (A1), with varying relaxation times between excitations.

recovered after a few minutes, confirming the reversible, short-lived nature of this synaptic mimicry. Such mechanisms are crucial for temporal filtering, sensory adaptation, and short-term information retention in biological circuits[68].

To conclude, we investigated four types of nanofluidic memristors, uncovering behaviors such as ion concentration polarization and surface charge inversion, and studied how temperature influences ion transport and memristive effects. All four behaviors coexist within the same device architecture (Fig. 2), driven by the interplay of surface charge, ion interactions, and nanochannel geometry. A minimal model was developed to explain these effects, supported by simulations matching experimental trends. Future nanoscale spectroscopic tools could further probe these mechanisms. Our findings highlight the versatility of nanofluidic memristors in mimicking biological functions like volatile/non-volatile memory, potentiation, depression, and short/long-term plasticity. Their tunable current or voltage responses enable advanced memory circuits and neuromorphic systems capable of complex, brain-like learning. We also demonstrated short-term depression leveraging system non-linearity. Future work could develop dynamic, reconfigurable logic for learning and responding to chemical or electrical stimuli.

## Methods

### Fabrication of nanochannel devices
We fabricate the nanochannel devices following the protocol detailed in ref. 69, illustrated as an example for $MoS_2$ nanocapillary devices with the flow chart (Supplementary Fig. 1). All 2D materials used for the top, spacer, and bottom layers of channels were prepared by mechanical exfoliation. First, we made a freestanding $SiN_x$/Si membrane (500 nm thickness) with a -3 × 25 μm hole using photolithography and dry etch by reactive ion etching. A $MoS_2$ (or hBN) flake was transferred onto the $SiN_x$ membrane as the bottom layer and etched from the backside to define the entrance of the electrolyte (hole side) (Supplementary Fig. 2a). Next, a graphene layer was patterned into a stripe shape with a width of 120 nm for the nanochannel, using electron beam lithography and dry etching. Then, a $MoS_2$ (or hBN) flake was transferred as a top layer onto the graphene layer (Supplementary Fig. 2b). The top-spacer stack was then transferred onto the bottom $MoS_2$ layer on the $SiN_x$ membrane in a perpendicular direction to the stripes, from the long-axis of the hole (Supplementary Fig. 2c). A metal layer (Cr/Au, 5/60 nm) was deposited onto the tri-crystal stack using photolithography and evaporation to protect the nanocapillary device (Supplementary Fig. 2d). Then from the top etching was done to remove top layer in regions other than that protected by the Au mask (Supplementary Fig. 2e). We used oxygen plasma to etch the graphene spacer on the hole in the $SiN_x$ membrane from the backside to clean the entrance. Finally, we annealed the nanocapillary device in a furnace with 10% $H_2$ gas at 300 °C for 3 h and then at 400 °C for 3 h to remove any polymer residues and improve the adhesion between the 2D materials. The resulting nanocapillary device was used for ion measurements.

### Characterization methods
Scanning electron microscope (SEM) images were obtained from a Zeiss ULTRA 55 (Zeiss, Germany). We measured the sample with an In-lens detector at an EHT of 3.00 kV. Atomic force microscope (AFM) images were obtained using a Dimension FastScan (Bruker, USA) in tapping mode.

### Electrochemical experiments
We used Ag/AgCl electrodes and an electrical source measurement meter, Keithley (2600 Series), equipped with a LabVIEW program to measure ionic currents/voltages with time-dependent voltage/current frequencies ranging between 0.2 and 165 mHz and amplitudes ranging from 0.05 to 1 V. The fluidic cell was first washed with distilled water, then increasing concentrations of electrolytes were added and measured. At the end of various concentration measurements and while changing the electrolyte, thorough washing of the cell using Milli-Q water was done. The whole electrochemical cell, holding the device and the measurement electrodes, was kept inside a Faraday cage to decrease the noise level.

The channel's instantaneous conductance, denoted as $G(t)$, was calculated from the current measurements using Ohm's law: $G(t) = I(t)/V(t)$. When needed, the pH of the solution was adjusted in a manner so that the concentration of the proton remained negligible compared to the main studied ions.

The experiments of temperature variation were conducted as follows: The heating process was carried out using an oven, which heated the entire fluidic cell. Subsequently, experiments were conducted at ambient temperature, with temperature monitoring throughout the measurements. As the heated cell was cooled, the temperature fluctuation was 1–2 °C during one voltage cycle. For the cooling experiment, the entire fluidic cell was placed in a freezer until a mixture of ice and liquid was formed. Similar to the heating experiment, these tests were also conducted at ambient temperature, and the actual temperature was measured using a multimeter that functioned as a thermometer with the use of a Type K thermocouple.

The inhibitory and excitatory action of zero-volt relaxation experiments was done by continuously applying a 1 V potential between Ag/AgCl electrodes across the channel membrane, then by relaxation at 0 V for a specific amount of time, followed by recording the current again while applying a 1 V potential. The difference in current before and just at the beginning of the second application of potential is recorded, echoing the excitatory and inhibitory post-synaptic currents behavior.

### Principal component analysis (PCA)
PCA been computed by writing the Python code where the features like molecular weight of salts (gram/mol), salt valency (mono 1, bi 2, tri 3), molar concentration of solution (M), height of nanochannel (nm), and materials (either hBN or $MoS_2$) in correspondence to four different type of loop style. Furthermore, we have encoded the material feature into numerical values with the help of the LabelEncoder technique from sk.learn pre-processing module in the scikit-learn library. This allows all features to have numerical values (which is a prerequisite for any numerical-based analysis). For consistency, all numerical features have been standardized with the help of the standard scaler technique from sk.learn pre-processing module in the scikit-learn library. This ensures all features have a 0 mean and a standard deviation of 1. Afterwards, we computed the PCA using a decomposition module from the Scikit-learn library. We have then automatically extracted the computed PCA values and weights (corresponding to individual features).

## Data availability
All the data supporting the conclusions of the study are presented within the main and supporting information. The raw experimental data of the main and supporting information are available in the Figshare repository, can be accessed at https://figshare.com/s/92119160b66c9653063e.

## Code availability
The input dataset, the computed principal component analysis (PCA) output result, and the Python script used for PCA analysis and simulations can be accessed through the Github repository: https://doi.org/10.5281/zenodo.15552632.

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

## Acknowledgements

B.R. acknowledges funding from the European Union's H2020 Framework Programme/ERC Starting Grant 852674 - AngstroCAP, Royal Society University Research Fellowship URF\R1\180127, URF\R\231008, Philip Leverhulme Prize PLP–2021-262, EPSRC new horizons grant EP/X019225/1. A.K. acknowledges EPSRC New Horizons grant (EP/V048112/1). B.R. and A.K. acknowledge EPSRC strategic equipment grant EP/W006502/1. B.R. and A.I. thank Prof Lydéric Bocquet for helpful discussion about surface charge inversion.

## Author contributions

B.R. designed, directed the project, and procured the funding. A.I. recognized and analyzed the two new memristive phenomena, led the measurements of the memristors, conducted their analysis, and provided all the data. G.H.N., A.K., K.V.S., Y.Y., R.S., and H.J. carried out the sample fabrication of Å-channel devices. G.H.N. and A.K. carried out the sample characterization. S.G. performed initial memory measurements. S.V.P. performed PCA analysis with inputs from B.R and A.I. A.L. performed simulations and developed a minimal model with inputs from B.R. and A.I. B.R. and A.I. wrote the manuscript with inputs from G.H.N., A.K., and all the authors contributed to discussions.

## Competing interests

The authors declare no competing interests.
