## [Transparent Peer Review file · Nature Communications]

Programmable memristors with two-dimensional nanofluidic channels

Corresponding Author: Professor Boya Radha

Version 0:

Reviewer comments:

Reviewer #1

(Remarks to the Author)

Gaining insight into the mechanism of ion transport in nanofluidics is crucial for both fundamental research and practical applications. The authors have reported that nanoconfined 2D slit-shaped channels exhibit typical memristive characteristics. In this manuscript, by rationally controlling and tuning the species and concentration of ions, they found that four distinct types of memristive loops can be controllably obtained within a single type of channel. Furthermore, time-dependent short-term depression within these nanochannels is discussed. The manuscript is well-written and organized; I think it could be publishable after addressing the following questions:

- 1) In Figure 2, it is challenging to differentiate between the colors of the circles; using different shapes along with color would enhance clarity.
- 2) In Figure S10, while observing rectification inversion in asymmetric solutions, I do not believe this inversion originates from surface charge inversion. Both cited references report inversions observed in symmetric solutions; thus, their mechanisms may differ significantly. The mechanism described in Langmuir (2012, 28, 2194–2199) may provide valuable insights into understanding the present phenomenon.
- 3) The authors noted varying rectification inversion factors for different monovalent salts; additional explanation should be provided in the manuscript.
- 4) In Figure S26, although the authors claim to have successfully mimicked action potentials in neurons using the as-prepared channel, my understanding suggests that the shape of these action potentials results from applied current waveforms rather than chemically controlled ion transport as seen in biological channels. Such shapes can typically be achieved across various channels by altering applied stimuli; therefore, caution should be exercised when demonstrating this point.
- 5) Several references closely related to this work should also be cited: Science 379:156–161 (2023); Chem. Asian J., 2022, 17(21): e202200682.
- 6) There are numerous typographical errors throughout the manuscript—such as 'Nb. 1175s' appearing in Figure 5b while its caption states '20 min.' Please ensure these are corrected meticulously.

Reviewer #2

(Remarks to the Author)

This paper reports an ionic memristor with four different kinds of transient hysteresis loops. It is interesting that the same memristor produces all 4 hysteresis loops with different ions and at different ionic strengths. Unfortunately, the authors provide a very cursory explanation of the 4 loops that is really not that enlightening.

Given that the current nanofluidic channel (so far as the ion current path is concerned) varies in cross section from the hole

to the outlet of the channel, it is geometrically similar to a conic pore or a funnel channel whose conductance varies along the current direction. It is well known that such channels exhibit rectification because of intrapore concentration enrichment and depletion (see, for example, Yan et al, J Phys Chem, 138, 044706(2015)). The hysteresis loops of channels of such geometry are also easy to understand--the current will go up upon voltage reversal in the high current polarity (intrachannel enrichment) and will go down on the low current polarity (intrachannel depletion). Because of the concentration polarization transient, the reverse path with a longer time will have higher enrichment or depletion. This kind of hysteresis has been observed and analyzed by Rosentsvit et al [J Chem Phys, 143, 224706(2015)]. Loops M1 and M3 of the current report belong to this category. That the enrichment/depletion polarity flips for the two cases is mostly likely due to charge inversion when Al^{3+} condenses on the negative charges at the lower ionic strength.

The M2 loop with a limiting current is most likely due to external polarization (Yossifon et al, Phys Rev Lett, 103, 154502(2009), see also Yan et al (2015)). A constant limiting current is only possible with external depletion at a pore tip. It cannot be established with intrachannel depletion. Yan et al gave a condition for the occurrence of this limiting current at a relatively high ionic strength when intrachannel resistance is negligible.

Hence, M1 to M3 loops are not that surprising or novel. The most interesting one is M4: the Wien loop. Also, unlike M1 to M3, the M4 loop is highly symmetric with respect to polarity change--there is no rectification effect. The authors suggest a Bjerrum pair polyelectrolyte is responsible for the increase in current during the back scan at both polarities (Fig. 3). There is no evidence that this rather speculative (bizarre ?) mechanism is true. Since this occurs with KCl at high ionic strength, charge inversion is probably not at play. I also do not understand why and how such polyelectrolyte can be formed.

I hence encourage the authors to cite the above papers for hysteresis loops M1 to M3 and focus the study on M4. Explain the symmetry and why there seems to enrichment at both polarities.

Reviewer #3

(Remarks to the Author)

The manuscript describes interesting work on fluidic memristors, based on previous architectures and models by the same group published in Science in 2023. The novelty of this work is the systematic exploration of different kinds of ions, slit heights, temperatures, and pH which enables to program the memristive characteristics to realise all 4 memristor types proposed in the literature. While the manuscript is potentially impactful, it is currently missing compelling explanations of the underlying phenomenology which remains mostly qualitative.

The main drawback of this work is the lack of a well-founded, possibly quantitative explanation of the rich and potentially impactful properties of these nanofluidic memristors. The invoked phenomena (except the Bjerrum pairs, see below) are all likely to be present, but discriminating between them requires quantitative insights and perhaps additional experimental evidence. a) The scatter of the data in Fig. 2 is quite broad and deserves more attention to explain trends, especially in relation to channel height that is not discussed in depth. The simple models proposed in Fig. 3 do not seem to unequivocally explain the data in Fig. 2. b) Quantitative models of concentration polarization and surface charge inversion could help disentangling the picture. c) The formation of Bjerrum pairs was also proposed in previous papers as a potential origin of memory in nanofluidic channels, but the same authors previously stated that "this process can only take place in thinner channels—those less than 2 nm in thickness—as Bjerrum pairs only exist under strong confinement". In a nutshell, the existence of Bjerrum pairs in nanoslits was not demonstrated and it is very unlikely to occur for taller channels. Therefore the subsequent reasonings invoking this phenomenon, including the interpretation of temperature trends, are not on firm grounds. d) The emergence of long term memory is not quantitatively compatible with the models considered in the text and in Fig. 3, see again the Science paper by the same authors.

On the simulated action potential section, there are some remarks on the protocol. The classical Hodgkin-Huxley model comprises two different memristors mimicking two different channels (K and Na) and a constant current is applied. The proposed protocol, instead, implies an ad-hoc current protocol which also has a threshold. It is therefore not clear whether the resulting all-or-none behaviour follows from the properties of the memristors or from the specific protocol adopted.

In conclusion, the paper contains promising results but, in the current form, does not provide sufficiently substantiated insights into the physical mechanisms at the basis of the memristive behaviour.

Version 1:

Reviewer comments:

Reviewer #1

(Remarks to the Author)

The revised manuscript entitled Programmable memristors with two-dimensional nanofluidic channels described a 2D nanofluidic memristor showing all 4 kinds of conductivity switch kinetics in different electrolyte environment. Undoubtedly, memristors with this programming ability was not reported in both nanofluidic systems and solid memristor devices. Generally, the manuscript was properly revised and all the raised problems were solved in the revised manuscript. This revised manuscript deserve publication in Nat. Commun., while I still have several minor questions which I hope can be taken into discussion.

It seems that when there is almost no charge effect, M4 Wien effect governed the observation while when charge effect governed the channel, the occurrence of M1, M2 or M3 is related to the surface-ion interaction intensity. For monovalent cations, the surface-ion interaction is limited, thus in most cases M4 was observed; And for multivalent cations, stronger interaction contributed to the concentration-dependent switch among M1, M2 and M3, right? Personally, I recommend the authors to give a brief conclusion on what interaction controlled the changes of memristor types among M1, M2, M3 and M4.

Again, in what cases, Wien effect could govern the whole system? It seems that there should be both strong charge inversion (surface-ion interaction) and strong ion-ion interaction in concentrated electrolyte. What parameter decided which one is the major effect?

One interesting thing is that the device showed LTP (M3), LTD(M1), STP(M4) and STD(M2) in the same voltage in different electrolyte, why the conductivity switch can be (cannot be) hold without bias voltages should be briefly discussed.

Page 11 Line 17 It seems that there is a typo error that the authors is giving description to Fig. 4C instead of 3D right ?

Reviewer #2

(Remarks to the Author)

In the revised manuscript, the authors have now clarified the asymmetric geometry of their nanochannel and do agree that the rectification phenomena they see are due to asymmetric external polarization at the two asymmetric entrances of their channel. This topic was first studied by Yossifon et al in their 2009 PRL paper (ref 56) and then by many others. Yossifon et al also used straight channels like the authors, not conic nanopores. The authors have made smaller nanochannels to allow for lower voltage and higher ionic strength. They have also used results from earlier reports to interpret their data. Nevertheless, it is not that novel.

The other claimed novelties of the paper are memory effect due to salt condensation/charge inversion, particularly monovalent salt concentration like K^+ , and hysteresis and memory due to Wien water dissociation.

The former charge inversion phenomenon has also been reported by many researchers, including the authors. The small nanochannels in the present study have, nevertheless, allowed for a larger class of charge inversion phenomena and the authors do offer a more comprehensive classification of the resulting different memory effects.

Their most novel results, however, are the M4 hysteresis by the Wien effect. The only known literature report of Wien hysteresis I am aware of are by Cheng and Chang (Biomicrofluidics, 5, 046502(2011); LabChip, 14, 979(2014)) with a theoretical explanation of the hysteresis due to concentration front propagations by Conroy et al (PRE, 86, 056104(2012)). The authors have now focused their manuscript on this less-studied hysteresis effect but they need to compare their results to these earlier studies.

If they can add these further revisions on the M4 Wien hysteresis, I believe the paper would then be acceptable for publication.

Reviewer #3

(Remarks to the Author)

Unfortunately, the feedback on this work cannot change from the previous submission: it contains promising results but the explanations remain qualitative. No simulations, theory, or models were added. Since the different reported regimes rely on a subtle balance between different phenomena, this is a serious drawback which prevents a broad and reproducible impact of these insights. Additionally, the explanation of loop type M4 based on Bjerrum pairs/polyelectrolytes is purely hypothetical as acknowledged by the authors; the statement that "the dielectric constant of the first water layers is decreased which allows this pairing near the channel walls" in the taller channels is also potentially interesting but unsubstantiated. Finally, it is again speculative the existence of "adsorption/desorption" at the surface at the heart of the reported "long-term memory" displayed by all channels. For these reasons, the publication of the manuscript is not recommended.

Version 2:

Reviewer comments:

Reviewer #1

(Remarks to the Author)

The revised manuscript entitled "Programmable memristors with two-dimensional nanofluidic channels" described a switchable memristor with memristive performance of 4 patterns, this switching kinetic discussed herein is of great importance in the field of nanofluidic. Generally all problems proposed are properly answered and I recommend to publish as it is.

(Remarks on code availability)

Reviewer #2

(Remarks to the Author)

The authors have conducted more experiments to demonstrate that their hysteresis is not due to the Wien effect of splitting water, whose hysteretic response has been reported in the papers they now cite. In particular, they show with DI water that they get the usual capacitive loop during voltage scans, without hysteresis. I believe this confirms that their observed hysteresis is due to splitting of Bjerrum pair and not water. It is a new phenomenon and I can now recommend this article for publication in Nature Communication. I believe it will attract many researchers to this field of memristor with transient hysteresis.

(Remarks on code availability)

Reviewer #3

(Remarks to the Author)

The authors have added a phenomenological theory that illustrates how the competition between different hypothetical microscopic mechanisms is compatible with the different memristive behaviours. Publication of the manuscript can now be recommended; I am looking forward to seeing in the future direct evidence of these interesting molecular mechanisms.

(Remarks on code availability)

Reply to reviewers

Programmable memristors with two-dimensional nanofluidic channels

Abdulghani Ismail^{1,2}, Gwang-Hyeon Nam^{1,2}, Yi You,^{1,2} Siddhi Vinayak Pandey,^{1,2} Hiran Jyothilal,^{1,2} Solleti Goutham^{1,2}, Ravalika Sajja^{1,2}, Ashok Keerthi^{2,3}, Boya Radha^{1,2*}

¹Department of Physics and Astronomy, School of Natural Sciences, The University of Manchester, Manchester M13 9PL, United Kingdom

²National Graphene Institute, The University of Manchester, Manchester M13 9PL, United Kingdom

³Department of Chemistry, School of Natural Sciences, The University of Manchester, Manchester M13 9PL, United Kingdom

* Correspondence to be addressed to: radha.boya@manchester.ac.uk

General comment

We sincerely thank the reviewers for their thorough assessment of our manuscript and for the insightful and constructive feedback. Their comments and suggestions have been invaluable in refining our work and enhancing the clarity and impact of the manuscript.

In response to the reviewers' remarks, we have made substantial modifications to the manuscript. This includes **clarifying our device geometry, expanding the discussion on the mechanism of each memory type, providing additional data to support our conclusions and discussing the novelty of the observed phenomena with respect to the existing literature**. We have also **revised the figures** in the main and in the supplementary for improved clarity, **added closely related references** and **corrected any typographical errors** noted. Additionally we performed principle component analysis to reduce the dimensionality of the variants affecting the observed memristive effects.

We are confident that the revisions address all the concerns raised and strengthen the manuscript significantly. We provide in the following paragraphs a detailed, point-by-point response to each comment of the three reviewers. We highlight our responses in **blue** and note the corresponding changes to the manuscript **in highlighted yellow**, for ease of reference.

Reviewer #1:

Gaining insight into the mechanism of ion transport in nanofluidics is crucial for both fundamental research and practical applications. The authors have reported that nanoconfined 2D slit-shaped channels exhibit typical memristive characteristics. In this manuscript, by rationally controlling and tuning the species and concentration of ions, they found that four distinct types of memristive loops can be controllably obtained within a single type of channel. Furthermore, time-dependent short-term depression within these nanochannels is discussed. The manuscript is well-written and organized; I think it could be publishable after addressing the following questions:

We thank the reviewer for the positive and constructive evaluation. We appreciate the time and effort dedicated to providing this feedback, which will undoubtedly help improve the clarity and rigor of our manuscript. Below, we address each of your comments:

Question 1.1

In Figure 2, it is challenging to differentiate between the colors of the circles; using different shapes along with color would enhance clarity.

We acknowledge the challenge relying solely on color differentiation. In this revised version, we have used different symbol shapes and colors to enhance the readability of the 3D graph and improved the overall appearance of the figure for easier interpretation.

Figure 2: 3D diagram and Principle component analysis depicting memristive phenomena at various conditions. The different memristor loop styles obtained from several A-B) hBN and C) MoS₂ devices. The x, y and z axis represent the salt type, salt concentration and the nanochannel height, respectively. MoS₂ devices showed crossing 2 style at low salt concentrations and for thick channels; Wien style at high concentration and thin

channels; saturation type mainly with trivalent aluminium chloride (or at very low concentrations of monovalent where the memory is mixed with the capacitive effects). A-B) The hBN devices showed crossing behaviour at high concentration, particularly for multivalent cations. The existence of two crossing types, one at low and other at high concentrations, occurs in several devices, which is attributed to charge inversion (discussed in figure 3); this effect is not seen in MoS₂ devices. The saturation style is limited to certain concentration range of aluminium salts and MnCl₂. D) PCA analysis of the measured devices (hBN, MoS₂) that reduce the dimensionality from 5 dimensions (salt type, valency, concentration, and channel height and channel material) to two principle components (PC1 and PC2) to enhance the visualization of clustering patterns and underlying phenomena.

Question 1.2

In Figure S10, while observing rectification inversion in asymmetric solutions, I do not believe this inversion originates from surface charge inversion. Both cited references report inversions observed in symmetric solutions; thus, their mechanisms may differ significantly. The mechanism described in **Langmuir (2012, 28, 2194–2199)** may provide valuable insights into understanding the present phenomenon.

Thank you for this suggestion, we now consider literature for asymmetric electrolytes to validate the inversion of rectification (figure S10 in previous version) in our observed M1 and M3 memristor effects. Two possible reasons for the rectification inversion could be hypothesized:

In the suggested article by the reviewer (Langmuir, 2012, 28, 2194–2199)[1], the rectification factor (RF) is influenced by the i) interplay between asymmetric ion transport and the diffusion gradient or ii) Surface charge inversion. We believe that both effects are at play, as emphasized now in the explanation and additional experiments provided in the SI of the article.

1. **Revised Literature Citations:** We have updated the citations to include relevant articles on charge inversion under asymmetric electrolyte conditions (concentration gradient). For example, He et al. (JACS, 2009) [2] demonstrated charge inversion by comparing monovalent (K⁺, Cl⁻) and bivalent (Ca²⁺ and Cl⁻) solutions (Figures 6 and 7 in the reference reproduced as Figure R1 below), where they observed SCI only using CaCl₂ solution. Additionally, Li et al (Nanoletters 2015) [3] predicted using simulations that the surface charge inversion (SCI) occurs in asymmetric conditions when bivalent ions rather than monovalent ions are used. Similar to the methodology we used in previous version figure S10, they kept a fixed concentration gradient and varied the concentration of electrolyte on both sides of a nanochannels (Figure 3a) (Figure R1). Li et al [4](ACS APPLIED MAT. Interfaces, 2019) reported various concentrations of electrolyte using fixed gradient on PET etched nanopore and correlated the RF variation with SCI. They observed this variation only in bi and trivalent salts and not using monovalent KCl solution.

[REDACTED]

Figure R1: Summary of the literature reports on surface charge inversion using monovalent and bivalent cations. Graphs adapted from [2-4].

We observed similar phenomena as described by Li et al. (Nano Letters, 2015) [3], with our methodology i.e., fixing the electrolyte gradient as 100 and varying the cis concentration, though the concentration range for charge inversion was higher (M vs mM), likely due to our specific system's 2D confinement in ultrathin slits. This shift in the concentration necessary for SCI towards a higher range between bulk and confined systems was also noted by Morikawa et al.. (Colloid and Interface Science Communications, 2022) [5] and attributed to a more highly charged surface in extended nanospace. Additionally, Li et al (ACS APPLIED MAT. Interfaces, 2019) [4] observed SCI in molar and higher millimolar ranges for bivalent and trivalent ions, respectively, in PET nanopores.

2. **Examining Ion Transport and Diffusion Gradient Effects:** To analyze the combined effects of asymmetric ion flow and the diffusion gradient, we replicated the conditions described in [1] (Langmuir 2012, 28, 2194–2199), inverting the concentration gradient while keeping the electrode orientation fixed. Under these conditions, the diffusion gradient opposes the electric field in the positive potential. This was tested with various monovalent and bivalent ions at low (mM) and high (M) concentrations. We observed variations in the rectification factor upon inversion of the concentration gradient, confirming the cumulative effects of diffusive ionic flow with geometric preferential flow. As noted below, we tested several conditions:

- a. **Case 1: Monovalent ions at low concentration range ($C_H=100$ mM, $C_L=1$ mM):** When the concentration is higher at the hole side (left graph), the RF (+/-) is higher than in symmetric conditions (centre graph Figure R2). This effect reverses when the concentration is higher at the device side (right graph Figure R2).

Figure R2: Diffusion gradient experiments (gradient = 100) at different directions at low NaCl concentrations in hBN device ($h=1.4$ nm).

- b. **Case 2, monovalent ions at high concentration range (e.g., 3M KCl, 6M NaCl, 10M LiCl):** Unlike the above scenario, the RF (+/-) becomes lower than in symmetric conditions with a gradient from the hole side, and vice versa (Figure R3). From this, we can deduce that the

variation is not due neither to geometric preferential passage nor to diffusive flow alone (the gradient direction and value are the same in both cases), thus the preferential passage has been changed by the high concentration and we believe this change is due to inversion of surface charge of the channel (which change the preferential flow as the main actor become anions rather than cations). To confirm so we tested other conditions:

Figure R3: Diffusion gradient experiments (Gradient =100) at different directions at high concentrations of monovalent electrolytes (KCl, NaCl and LiCl) in hBN device ($h=1.4$ nm).

- c. **Case 3: Bivalent salt at low concentration range ($C_H= 100$ mM).** Here we observed inverse to case 1 of monovalent and similar to case 2. When the concentration gradient is from the hole side (and the positive polarity is always from the hole side), the RF (+/-) become lower (Figure R4).

Figure R4: Diffusion gradient experiments (Gradient =100) at different directions at low concentrations of bivalent cation electrolytes (MgCl_2 and MnCl_2) in hBN device ($h=1.4$ nm).

- d. **Case 4: Bivalent salt at high concentration range ($C_H = 5$ M).** Similar to Case 3, the RF remains consistent across high and low concentrations, suggesting surface charge inversion at both ranges for bivalent ions but only at high concentrations for monovalent ions (Figure R5). The inversion of surface charge by monovalent ions has not been observed previously in artificial nanochannels (He, JACS 2008; Li et al., ACS Applied Materials & Interfaces, 2019; Li, Nano Letters 2015; Han et al., Nano Letters 2024; Van der Heyden 2006) [2-4, 6, 7]. Rezai et al [8] (PhysChemChemPhys, 2018) estimated that surface charge inversion (SCI) would occur only at molar salt concentrations. While surface charge inversion by monovalent ions has been observed in biological nanopores [9] (Lopez et al., Electrochemistry Communications, 2014) and colloid systems [10] (Calero et al., JACS, 2011), it has not been reported in artificial nanochannels.

Figure R5: Diffusion gradient experiments (Gradient =100) at different directions at high concentrations of bivalent cation electrolyte (MnCl_2) in hBN device ($h=1.4$ nm).

- e. **Case 5: Bivalent ions (e.g., MgCl_2) at high concentration (5M) with varying concentration gradient values:** We chose these conditions to ensure surface charge inversion. We observed that the rectification factor (RF) increases proportionally with the diffusion gradient, confirming the gradient effect. This clearly demonstrates the effect mentioned by the reviewer, where the RF can vary under asymmetric conditions due to the gradient itself. However, we hope based on the other contrasting conditions (cases 1-4) are sufficiently clear to show that both effects are indeed occurring

Figure R6: Diffusion gradient experiments (Gradients 0, 10, 100, and 500) from the device to the hole side at high concentrations of MgCl_2 in hBN device ($h=1.4\text{ nm}$).

3. **Symmetric Condition Results:** We also observed similar trends in rectification factor and memory style across different monovalent electrolyte concentrations under symmetric conditions (without changing electrode positions; see Figure S16). This variation depends on the monovalent cation type, likely due to differential cation binding to the channel walls and/or collective effect, resulting in unique rectification factor variations (Figure S16) (see discussion in main about SCI and section 6 in SI). This symmetric ion rectification effect was also noted by Xiong et al. [11] (Science China Chemistry, 2019) in polymer-functionalized nanopipettes, without exhibiting memristive effects—highlighting the novelty of our confined system.

In summary, we observed a phenomenon anticipated in prior research, yet our specific device structure allowed us to observe it more prominently, including with monovalent ions.

Question 1.3

The authors noted varying rectification inversion factors for different monovalent salts; additional explanation should be provided in the manuscript.

We have added a graph showing the rectification factor (RF) variation for three different salts as a function of concentration (Figure S16 in SI and Figure R7). The figures indicate that RF variation is salt-dependent across the tested monovalent salts. This dependence may partially stem from variations in surface charge density and/or the concentrations required for surface charge inversion, which differ based on the specific cations involved [3, 6, 8]. Additionally, this variation could be linked to ion-ion correlations, where both the cations and their interactions with anions play a role [12] (M. de Vos, *Advances in Colloid and Interface Science*, 2019).

This salt dependence in asymmetric conditions for different monovalent and bivalent salts, as discussed in the previous response, highlights that the RF sensitivity varies according to cation type, even under identical gradients.

Figure R7: Effect of increasing electrolyte's concentration on the rectification of current. Equimolar concentration conductance-voltage characteristic of KCl; and NaCl electrolyte using hBN pristine device ($h = 1.3$ nm) at increasing concentrations indicated above each GV characteristic curve.

The text of surface charge inversion is modified to the following on pages 8 and 10

This indicate that the main charge carrier varies depending on the concentration range - it is the anion at high concentration and the cation at low concentration (see discussion in supporting information, section 6). This effect was similar at high and low concentration ranges in other monovalent cations (K^+ and Li^+), while for bivalent cations (Mn^{2+} and Mg^{2+}) their low and high concentration range show only one behaviour akin to monovalent ions' high concentration range (Figure S12-S13). This indicated that the main charge carrier is the same at both concentrations for bivalent cations. We inverted the concentration gradient, and confirm that it is the main charge carrier that is inducing the variation in the RF and not the direction of the diffusive ion flow (Figure S12, S13). Additionally, we varied the strength of the gradient by increasing the difference in the concentrations on both side of the channel and observing the variation of the RF (Figure S14 and see the discussion in section 6, SI).

-The phenomena of surface charge inversion is illustrated in Figure 3. **SCI occurs when a multivalent ion is attracted to the channel's negatively charged surface, resulting in overscreening and a reversal of surface potential from negative to positive. This phenomenon arises from ions transverse correlation with the wall's surface charges and/or lateral correlation with each other (as described by strongly correlated liquid theory) at high salt concentrations [13].** This leads to **dependence of charge inversion upon several factors, such as the discrete nature of charged surface sites, counterion valency, counter-ion to co-ion size ratio, surface charge density, nanochannel diameter, solvent dielectric constant, and $\text{pH}^{32-35, 29}$.** Our experimental findings reveal charge inversion with bivalent **and trivalent** ions as well as monovalent ions when coupled with chloride co-ions. Testing cations with sulfate did not result in crossing type memory (only Wien) (Figure S24). This emphasizes the significant role of ion-ion interactions in SCI. **This observation of SCI in synthetic nanochannels with monovalent salts is novel; it has not been previously reported in synthetic nanochannels (He, JACS, 2008; Li et al., ACS Applied Materials & Interfaces, 2019; Li, Nano Letters, 2015; Han et al., Nano Letters, 2024; Van der Heyden, 2006), though it was predicted by simulation (Rezai, PhysChemChemPhys, 2018).** The unique 2D confinement of our nanochannels likely enables this inversion to occur with monovalent cations, resulting in a hysteresis memory effect. This could also explain previous observations of charge inversion in biological nanochannels at high monovalent salt concentrations (Lopez et al., Electrochemistry Communications, 2014). Furthermore, beyond cation valency and co-ion type, other parameters, such as the choice of monovalent cations, contribute to the nuanced nature of charge inversion. Notably, lithium chloride (LiCl) and sodium chloride (NaCl) exhibit a higher degree of rectification factor inversion compared to potassium chloride (KCl) (Figure S16). This was revealed by symmetric and gradient comparison of the RF inversion (Figure S12 and S16).

Figure R8: Effect of increasing LiCl concentration on the rectification of current in MoS₂ device. Equimolar concentration conductance-voltage characteristic of LiCl electrolyte using MoS₂ pristine device ($h = 0.7$ nm) at increasing concentrations from 0.01M to 3M indicated above each GV characteristic curve. The ratio of the positive to negative conductance varied with increasing the concentration signifying simultaneous charge inversion where the proportion of cations and anions in the electrolyte might vary..

Question 1.4

In Figure S26, although the authors claim to have successfully mimicked action potentials in neurons using the as-prepared channel, my understanding suggests that the shape of these action potentials results from applied current waveforms rather than chemically controlled ion transport as seen in

biological channels. Such shapes can typically be achieved across various channels by altering applied stimuli; therefore, caution should be exercised when demonstrating this point.

We appreciate the reviewer’s insight on this matter. We also tested samples with microholes without confinement, and they exhibit resistive rather than memristive behavior, applying the same program described in the SI result in a voltage responses that mirror the current variation (black and red lines in the figure R9):

There was a consistent transformation factor between voltage and current, representing resistance in the micropore experiments (Figure R9, microhole). However, in the nanochannels, this factor was asymmetric and variable due to memristance (Figure R9 Nanochannel). Our input is indeed variable, and the threshold does not have a physical meaning for the nanochannel but it was programmed to mimic what is present in biological channels where there is a threshold limit that depends on the type of ion-channel. So we set the hypothetical value of 100 mV as threshold for our nanochannels in the algorithm, above which current sweeps continue and below which applied current becomes zero.

To avoid any potential confusion, we have removed Figure S26 from the manuscript, along with the corresponding text, in response to concerns raised by reviewers 1 and 3.

Figure R9: variation of the potential variation across the nanochannel while applying current program under confined system and in microhole device. The figure shows that the confined nanochannels shows non-linear variation of the voltage vs current (memristive behaviour), however the microhole shows perfect correlation

Reviewer 1 comments

between the applied current and the measured voltage (resistive behaviour). The general current protocol used in this kind of experiments is illustrated in the flowchart (Nb: the current sweep values were adjusted depending on the used device).

Question 1.5

Several references closely related to this work should also be cited: Science 379:156–161 (2023); Chem. Asian J., 2022, 17(21): e202200682.

We thank the reviewer for this comment. We revised the references and enriched them with recent relevant references including the ones mentioned here

Question 1.6

There are numerous typographical errors throughout the manuscript—such as 'Nb. 1175s' appearing in Figure 5b while its caption states '20 min.' Please ensure these are corrected meticulously. We revised the thoroughly the manuscript to correct the typological errors, thank you.

Reviewer #2:

This paper reports an ionic memristor with four different kinds of transient hysteresis loops. It is interesting that the same memristor produces all 4 hysteresis loops with different ions and at different ionic strengths. Unfortunately, the authors provide a very cursory explanation of the 4 loops that is really not that enlightening.

We thank Reviewer 2 for the insightful comments and for highlighting the need for a more detailed explanation of the four memory hysteresis loops observed in our ionic memristor. In response to this feedback, we have expanded our analysis and discussion of the mechanisms behind these hysteresis loops, highlighting their key differences from existing literature and emphasizing their novelty. We enhanced all the figures and answer all the comments as below.

Question 2.1

Given that the current nanofluidic channel (so far as the ion current path is concerned) varies in cross section from the hole to the outlet of the channel, it is geometrically similar to a conic pore or a funnel channel whose conductance varies along the current direction. It is well known that such channels exhibit rectification because of intrapore concentration enrichment and depletion (see, for example, Yan et al, J Phys Chem, 138, 044706(2015)). The hysteresis loops of channels of such geometry are also easy to understand--the current will go up upon voltage reversal in the high current polarity (intrachannel enrichment) and will go down on the low current polarity (intrachannel depletion). Because of the concentration polarization transient, the reverse path with a longer time will have higher enrichment or depletion. This kind of hysteresis has been observed and analyzed by Rosentsvit et al [J Chem Phys, 143, 224706(2015)]. Loops M1 and M3 of the current report belong to this category. Hence, M1 to M3 loops are not that surprising or novel.

We thank the reviewer for this comment. We recognize the importance of clarifying our 2D nanochannel geometry and discussing similarities and differences with other channel types (as shown in Figure R10 and Figure S1 in Supplementary information).

Figure R10: Cross-sectional Illustration clarifying the geometry of our 2D nanochannels compared to Funnel-shaped nanochannels.

Reviewer 2 comments

Our 2D nanochannels are symmetrical along the ion path plane, resembling a parallelepiped with a constant cross-section, unlike funnel-shaped channels that vary in width from base to tip. These slit-like nanochannels allow free ion movement in two dimensions (length and width) while restricting it in height, ranging from 0.68 to few nm. In contrast, conic pores have asymmetrical confinement: their tip diameter typically ranges from 2–5 nm, while the base can reach hundreds of nanometers. In conical nanopores, the strongest confinement is at only at the tip entrance. This difference in geometric symmetry as well as size/dimensions creates distinct ion confinement levels, with much stronger confinement in our nanochannels.

The variation in rectification between polarities in our system is attributed to access resistance between the hole and device sides of the channel [14] (Science, 2023), rather than to the nanochannel structure itself, as in funnel-shaped nanochannels. In this revised version, we have expanded on rectification variation by including **additional experiments** on coupling asymmetric entrance effect with concentration gradients (Figures S12, S13, S14) (Also please see comment 1.2 to Reviewer 1). Our nanochannels display novel phenomena not previously reported in synthetic nanochannels:

- (1) Observation of memory behavior, reflected by pinched hysteresis that varies with frequency and retains memory over different timescales, discussed throughout the manuscript.
- (2) Surface charge inversion in monovalent cations including Na^+ , K^+ and Li^+ .

The rectification effects in the studies referenced by reviewer, explain asymmetry in current but do not account for memory effects, where the system "remembers" past input due to ionic interactions.

The innovative part is not only observing rectification or variation of rectification, rather is to observe the memory alongside with rectification. The memory necessitates variation in the kinetics of adsorption/desorption of the counterion inside the nanochannels, in addition to rectification. Indeed the concentration used are three orders lower and they do not observe this RF when they use our concentration range reflecting wide discrepancies between the studies making it hard to draw parallels. To the best of our knowledge, the memory behaviours (M1-M4) that we report with our 2D nanochannels were not reported before to coexist in any article. A very recent article [15](Ling, 2024, JACS) showed how surface charge inversion can invert the memory akin to M1 and M3 however they used two different systems of nanopipets functionalized with different surface charge molecules (Tannic acid and APTES). We are contributing an innovative approach to achieve four different types of ionic memory along with an understanding of the mechanisms in this article.

A new figure S1 is added to the supporting information.

Figure S1: Schematic of the geometry of the nanocapillary device. A) 2D cross-sectional view of the device illustrates the dimensions and walls of the nanochannels. The device comprises ~200 parallel nanochannels,

each shaped as a parallelepiped with a length in the micrometer range (top inset). The spacer's height determines the nanochannel height (ranging from 0.68 to 100 nm), while the distance between the graphene nanoribbons sets the nanochannel width at ~ 120 nm. B) The ion flow direction from channel side to the hole side. The flow can be either from hole side or channel side depending on the gradient and the driving voltage. C) Simplified depiction of the same flow direction as in B, is used in some of the figures.

Question 2.2

Loops M1 and M3 of the current report belong to this category. That the enrichment/depletion polarity flips for the two cases is mostly likely due to charge inversion when Al^{3+} condenses on the negative charges at the lower ionic strength.

We thank the reviewer for acknowledging charge inversion as a mechanism for the observed memory effects M1 and M3. We would like to emphasize that this behavior corresponding to the M1 effect is not exclusive to Al^{3+} ions; we have observed it with divalent ions and even with monovalent ions in hBN nanochannels. This is an important point, which we have highlighted in the revised version with additional experiments. Notably, M1 was not observed in similar 2D nanochannels made from MoS_2 , despite the fact that variation of rectification factor is observed, underscoring the significance of both geometry and material in these phenomena (see figure R11 and Figure S17 in SI).

Figure R11: Effect of increasing LiCl concentration on the current rectification in MoS_2 device. Conductance-voltage (GV) characteristics of LiCl electrolyte at equimolar concentrations, ranging from 0.01 M to 3 M, using MoS_2 channels ($h = 0.7$ nm). The ratio of positive to negative conductance changes with increasing concentration, suggesting simultaneous charge inversion, due to variations in the relative proportions of cations and anions in the electrolyte. LiCl 0.01M shows M3 (crossing effect) while for concentrations higher than 0.1M, all the observed memristive effects were Wien effect (M4).

Question 2.3

The M2 loop with a limiting current is most likely due to external polarization (Yossifon et al, Phys Rev Lett, 103, 154502(2009), see also Yan et al (2015)). A constant limiting current is only possible with external depletion at a pore tip. It cannot be established with intrachannel depletion. Yan et al gave a condition for the occurrence of this limiting current at a relatively high ionic strength when intrachannel resistance is negligible.

We thank the reviewer for confirming our explanation of the M2 loop style, attributed to external polarization. However, we would like to emphasize the originality of the phenomena observed in our nanochannels, which differentiates our work from previous reports, despite certain similarities.

Reviewer 2 comments

- We observe a ‘memory’ effect that varies with frequency of the applied voltage. In previous reports, although hysteresis is present, it is more of a capacitive curve rather than pinched memory loop.
- Our setup achieves ion concentration polarization (ICP) regime at much lower voltages than previously reported, suggesting potential energy savings for applications as a memristor.
- We are operating at significantly higher concentrations than typically reported in the literature (up to three orders of magnitude higher). For example, Yossifon et al. [16](PRL, 2009) found that their nanoslit device (190 nm, 100 μm , 450 μm) did not exhibit this ICP regime or rectification above 1 mM, and they used voltages up to 40V, which is at least an order of magnitude higher than in our current work. Interestingly, in our study with narrow confinement heights of the slit-channels to only few nanometers, the memristive effect occurs at low voltage, does not invert the rectification factor, and is independent of rectification factor changes. Thus, M2 and M4, which are type II non-self-crossing loops, do not rely on rectification factors as a basis.

We believe that these phenomena are observed under our specific concentrations and conditions, especially given we have array of parallel nanochannels which are highly selective in the conditions where M2 is observed, with access resistance playing a role. Theoretical models proposed recently by Cao et al. [17] (Advanced functional materials 2024) on ion concentration polarization in nanopore arrays are now cited in our work, which would help understanding the phenomena although they did not observe memory effects.

Question 2.4

The most interesting one is M4: the Wien loop. Also, unlike M1 to M3, the M4 loop is highly symmetric with respect to polarity change--there is no rectification effect. The authors suggest a Bjerrum pair polyelectrolyte is responsible for the increase in current during the back scan at both polarities (Fig. 3). There is no evidence that this rather speculative (bizarre ?) mechanism is true. Since this occurs with KCl at high ionic strength, charge inversion is probably not at play. I also do not understand why and how such polyelectrolyte can be formed. I hence encourage the authors to cite the above papers for hysteresis loops M1 to M3 and focus the study on M4. Explain the symmetry and why there seems to enrichment at both polarities.

We thank the reviewer for their interest in the M4 mechanism. The polyelectrolyte Bjerrum pair non-linear theory was predicted by MD simulations by Bocquet & co-workers (Science, 2021), and similar 2D nanoslits were experimentally demonstrated for ionic memory in our recent work [14](Science, 2023). Additionally, recent literature on molecular dynamics simulations have suggested the formation of monolayer hydrated salts [18, 19](Zhao, JACS, 2022; Zhao, Nat. Comm., 2021) and polyelectrolytes inside 2D nanoslits [20](Liang, Chem. Sci., 2024), indicating their possible existence even without an applied electric field.

We recognize that further experimental validation, using complementary techniques beyond electrical characterization, is necessary to confirm the presence of Bjerrum pairs or polyelectrolytes. However, the alignment of these simulations with our experimental electrical data supports the existence of this mechanism. Other concurrent mechanisms, such as ionic Coulomb blockade [21](Feng, Nat. Mat., 2016), surface-charge ionic blockade [22](Xie, Nanoscale, 2023), or hydrophobic gating[22], can likely be ruled out as our experimental conditions differ significantly from those under which these effects have been observed. The lack of experimental characterization methods is due to the extreme

confinement and the requirement for channel potential application during characterization in the liquid state.

The Wien effect, which can be symmetric or asymmetric depending on influencing factors, produces an apparent memory effect distinct from the isolated memristive effects theoretically described in this article. Additional data about the variation of rectification factor in monovalent salts presenting Wien effect are added in Figure S16 and S17. The coexistence and partial separability of these effects under specific conditions are, in our view, a unique strength of this study, confirming the presence of four mechanisms.

We also wish to emphasize that the saturation M2 loop can be either symmetric or asymmetric depending on the experimental conditions. We have added more experiments in this version to further discuss the symmetry observed in M4 (Figure R12 and S26), where three subtypes of external polarization M2 could be seen experimentally including negative differential resistance (NDR), limiting current, and overlimiting current (extended space charge) (see figure below).

- In NDR the current decreases as the voltage increases, this has several explanations. For example Siwy et al. [23] (2006, nanoletters) explained it in asymmetric nanopores by the interaction of ions with the negatively charged pore walls, which reverses the direction of rectification. The bivalent ion binding and unbinding induce fluctuations in the potential profile, driving ionic transport via a flashing ratchet mechanism. At specific ionic concentrations and voltages, these dynamics reduce ion flow at higher voltages, resulting the observed NDR effect. Lin et al. [24] (ACS Applied Materials and interfaces, 2019), attributed to electrodiffusioosmosis, where competing electroosmotic and diffusioosmotic flows alter ionic conductivity near the pore opening. At high voltages, electroosmosis dominates, driving low-conductivity solution into the pore and causing a decrease in ionic current. This NDR effect depends on the salinity gradient, pH-regulated surface charges, and the geometry of the mesopores. Ramirez et al. [24] (ACS Applied Materials and interfaces 2021) attributed the NDR behaviour observed in conical nanopores immersed in KF solutions at low concentrations to the accumulation and interaction of fluoride ions with negatively charged pore walls, which alters ionic conduction near the pore tip, and is influenced by ionic concentrations, pore charge density, and solution dynamics. In our current experiments NDR was observed mostly at low frequencies and we could attribute it to the interaction of ions with the channel walls as the electroosmotic behaviour is negligible in ultraconfined nanochannels [25, 26]. More recently, Yang et al. [27] further expanded the understanding of NDR in conical nanopores by demonstrating its emergence from the redistribution of ionic charges during hysteretic and rectified transport. They showed that NDR arises fundamentally from the interaction of ion concentration polarization and electroosmotic flow, which generates a dynamic enrichment and depletion of ions near the nanopore (or nanopipet) tip. By applying a triangular potential waveform, they revealed deterministic and chaotic state-switching behaviours. Their findings highlight that NDR can be tuned by manipulating ionic strength, nanopore geometry, and surface charge density, offering a simplified yet robust mechanism for generating high-order complexity in iontronic devices.
- In limiting current type, there exist two regions, an initial Ohmic region, a limiting current plateau with no slope, while in extended space charge there exist a third region, an overlimiting region with increased slope. We have observed experimentally the limiting current region in most cases at higher concentrations while the overlimiting current was observed in some cases at low ionic concentrations (10 mM). The explanation of these phenomena could be the concentration polarization where the concentration gradient is formed due to ion depletion at

Reviewer 2 comments

the entrance of the nanochannel resulting in a fixed amount of ions, and thus current, passing to the nanochannel.

- The Overlimiting current type is caused by the extreme depletion of ions at the channel entrance and the subsequent formation at low concentration of extended polarized space-charge layers where vortices and spontaneous convective mixing at high voltages near the channel entrance result in the increase of the slope in region 3 [28]

Figure R12: Behaviour of M2 saturation memristor observed in nanochannels. Three different types of saturation memristor A) negative differential resistance, B) limiting current, and C) overlimiting current were observed, the experimental conditions are written on each curve.

Reviewer #3:

The manuscript describes interesting work on fluidic memristors, based on previous architectures and models by the same group published in Science in 2023. The novelty of this work is the systematic exploration of different kinds of ions, slit heights, temperatures, and pH which enables to program the memristive characteristics to realise all 4 memristor types proposed in the literature. While the manuscript is potentially impactful, it is currently missing compelling explanations of the underlying phenomenology, which remains mostly qualitative. The main drawback of this work is the lack of a well-founded, possibly quantitative explanation of the rich and potentially impactful properties of these nanofluidic memristors. The invoked phenomena (except the Bjerrum pairs, see below) are all likely to be present, but discriminating between them requires quantitative insights and perhaps additional experimental evidence.

We thank the reviewer for confirming the novelty and significance of our work. We have addressed the issues related to the clarity of the mechanisms by enhancing the discussion and incorporating additional cross-referencing to the literature, highlighting unique aspects of our confined system that allow these phenomena to be distinguished from one another. We revised the figures and added experimental data to emphasize the mechanisms, performed principle components analysis to reduce the dimensionality and incorporate more parameters within the same analysis.

In this article, we aimed to deepen the understanding of these phenomena by controlling each parameter individually and experimentally wherever feasible. We added more experiments to investigate the charge inversion mechanisms and the interaction of ions with the channel walls. The polyelectrolyte Bjerrum pair non-linear theory was predicted by MD simulations by Bocquet et al. [29](Science, 2021), and non-linearity in similar 2D nanoslits were experimentally demonstrated for ionic memory in our recent work [14](Science, 2023). Additionally, recent literature on molecular dynamics simulations have suggested the formation of monolayer hydrated salts [18, 19] (Zhao, JACS, 2022; Zhao, Nat. Comm., 2021) and polyelectrolytes inside 2D nanoslits [20] (Liang, Chem. Sci., 2024), indicating their possible existence even without an applied electric field. We recognize that further experimental validation, using complementary techniques beyond electrical characterization, is necessary to confirm the presence of Bjerrum pairs or polyelectrolytes. However, the alignment of these simulations with our experimental electrical data encourages this mechanism. Other concurrent mechanisms, as ionic Coulomb blockade [21](Feng, Nat. Mat., 2016), surface-charge ionic blockade [22](Xie, Nanoscale, 2023), or hydrophobic gating[22], can likely be ruled out as our experimental conditions differ significantly from those under which these effects have been observed. The lack of experimental characterization methods is due to the extreme confinement and the requirement for channel potential application during characterization in the liquid state.

We modified the text as follows in main text on page 13-14

The explanation of Wien mechanism is challenging; it was first simulated by Robin et al [29] where Bjerrum pairs were predicted to be observed in the absence of any applied voltage, that transform to polyelectrolyte under electric field in 2D confined nanochannels. Following this, the Wien style memory was shown experimentally in MoS₂ pristine 2D nanochannels and the current-voltage characteristics of the experiment matched the simulation, albeit with exceptionally long timescales [30]. We explained this long-term memory (minutes to hours), by taking into account the slow effective transport due to pairing/unpairing mechanism of 2D confined ions (and by ‘stop and go’ mechanism of adsorption/desorption in case of M3 loop in activated carbon nanochannels). Although the existence of polyelectrolyte and Bjerrum pairs is not yet proven experimentally (e.g., electron microscopy or with spectroscopic methods), but the alignment of the simulations with our experimental electrical data supports this mechanism.[20] Other concurrent mechanisms, such as ionic Coulomb blockade [21],

surface-charge ionic blockade [22], or hydrophobic gating [31], can likely be ruled out as our experimental conditions differ significantly from those under which these effects have been observed. The nanoscale confinement and the simultaneous potential application during characterization in the liquid state pose technical challenges for experimental identification of Bjerrum pairs/polyelectrolyte transformations.

Additionally, we showed that these two effects (M3 and M4) can exist in hBN devices as well as in MoS₂ devices and we show how to switch between these two styles using the same channel device by modifying the concentration and pH (figure S15)

Question 3.1

The scatter of the data in Fig. 2 is quite broad and deserves more attention to explain trends, especially in relation to channel height that is not discussed in depth. The simple models proposed in Fig. 3 do not seem to unequivocally explain the data in Fig. 2.

We thank the reviewer for this comment that gave us the opportunity to clarify more about the effect of the height. As the 3D graph might be harder to read in the height effect, we emphasize the 2D figure in SI, corresponding to KCl that describe the effect of the height on the on/off conductance ratio and on the type of the memory loop with a corresponding text discussing the height effect when using KCl (section 7 SI). We measured several devices with varying heights and added the effect of height using AlCl₃ salt in hBN and MoS₂ devices (figure S20 and S21 in SI, Figures R13-14), and a paragraph in the main text to discuss the height effect as below.

Supporting section S8

Similarly, in the case of AlCl₃ electrolyte, the lower height (more confined nanochannels) show better loop with saturation (M2) memristive effect (Figure S21). This effect decreases with the height of the nanochannel but it can persist to heights upto 20 nm (vs 10 nm for KCl Wien effect) (Figure S21). Using very thick nanochannels the hysteresis area decreases a lot and there is a tendency toward obtaining an ohmic behaviour. This reflects in both cases, using KCl or AlCl₃, the importance of 2D confinement in obtaining memristive effect.

The results in Figure S20 provide summary of how channel height and electrolyte concentration affect the memristive effect in nanochannels with MoS₂ and hBN walls. The electrolyte tested here contains the trivalent Al³⁺ cation. Notably, Both in MoS₂ and hBN devices, the saturation style (indicated by black symbols) occurs at low concentrations ($\leq 0.1M$) across all channel heights and other types of loops appear at high concentrations for all tested heights. This phenomenon could be attributed to the inherent mechanism of the saturation memristive style, i.e., concentration polarization.

In MoS₂ devices, Wien effect occurs in low channel heights (<10 nm) and high salt concentrations (>1M). At higher heights, crossing 2 effect exist but with very small loop areas. This could be explained by the decrease in selectivity of the nanochannels which does not permit the conditions for external polarization to occur and thus the adsorption/desorption occurs. It is important to note that as the concentration and height of the channel increase, selectivity decreases, leading to a diminished saturation effect and to the decrease of the area of the loop (Figure S21).

In hBN devices, the emergence of the crossing-1 mechanism (green down triangles) occurs at low heights and intermediate to high concentrations. This phenomenon is due to the surface charge inversion mechanism that is pronounced with multivalent ions (here Al³⁺), where the negative surface charge of the channel turns positive through the adsorption of Al³⁺ ions, and the hysteresis mechanism

results from the difference in kinetic mechanisms of adsorption and desorption. The conditions for this memristor are – thin channels for the selectivity and rectification. Wien effect occurred at heights > 3nm which involves both cations and anions. In even thicker channels, the memristive effect disappears and tend more toward small crossing 2 effect.

Figure R13: Memristor dependence on AlCl₃ electrolyte concentration, height of the nanochannel and the material of the nanochannel. The different observed Memristor effects (Saturation, Wien, Crossing 1 and crossing 2) are represented by different colors and shapes. Alternating voltage range is between ± 1 V.

Figure R14: Memristor dependence on height using AlCl₃ as electrolyte. Current-voltage characteristics showing the decrease of the memristive loop for increasing memristor heights. Alternating voltage range is between ± 1 V

The main text is modified as follows on page 14-15:

The height of the nanochannel is fundamental to achieving 2D confinement and has a significant effect on all the memory styles and their existence. All thick nanochannels (>10 nm), exhibited very small pinched loop tending to an ohmic behavior (Figure S18-19, S21). For Type I self-crossing memristors (M1 and M3), this phenomenon can be attributed to the absence of intrachannel

resistance limitation, which are essential for observing different conductance states and current rectification. In the case of Wien memristors, while simulations by Robin et al.[29] suggested that Bjerrum pairs, and consequently M4 memristors, should only exist under confinement conditions of up to 2 nm, our observations extended their presence to channels as wide as 10 nm. This discrepancy may be due to the formation of Bjerrum pairs in the water layers adjacent to the channel walls. The reduced dielectric constant in these water layers could facilitate this pairing near the walls. The M2 saturation memristor showed non negligible loop area for thick channels up to 20 nm (see SI figure S9 and S21). This behavior relates to its underlying mechanism, which involves external polarization. Specifically, the depletion zone at the nanochannel entry persists, particularly at low ionic concentrations ($C < 100$ mM). At higher concentrations ($C > 1$ M), the pinched hysteresis diminishes and eventually disappears. The effect of nanochannel height is also evident in the behavior of the crossing 1 memristor, which relies on surface charge inversion. This memristor is primarily observed in nanochannels with heights below 2 nm at high concentrations of Al^{3+} . Notably, the effect becomes more pronounced at lower concentrations in highly confined nanochannels. The relationship between confinement and the concentration required to observe surface charge inversion has been discussed in the literature [4, 5], with SCI occurring at low concentrations in highly confined nanochannels.

Question 3.2

Quantitative models of concentration polarization and surface charge inversion could help disentangling the picture.

We added several experiments to demonstrate surface charge inversion in monovalent and bivalent salts using symmetric and asymmetric electrolyte concentration conditions (See SI figures 12, 13, 14, 16, 17, 25). This permitted us to understand the concentration ranges of surface charge inversion depending on ions. Indeed, we cross-referenced our findings with theories and models present in literature. While these studies explored similar issues, they utilized different nanochannel geometries and did not observe memory effects. We emphasized that our results not only reveal memory effects but also demonstrate charge inversion with monovalent cations, a phenomenon typically observed only in biological nanochannels.

Recognizing that multiple factors influence the memory mechanism—and that a balance among these factors likely exists—we performed principal component analysis (PCA) to better understand and visualize the correlations between various parameters. These parameters included salt molecular weight, valency, concentration, the height of the nanochannel, and its material. The PCA revealed additional insights by incorporating more parameters into the analysis. While the data could not be completely segregated, clusters with similar properties emerged. These findings could be further refined using machine learning techniques to optimize and enhance the observed memristive effects.

We added the PCA analysis to main figure 2 and the following phrases in page 3:

Additionally, a complementary Principal Component Analysis (PCA) was performed to visualize trends in loop styles based on multiple parameters after reducing their dimensionality to a 2D graph (Figure 2D). The analysis revealed clustering of memristive effects, although complete segregation was not observed. This indicates that similar characteristics are shared among memristor types, despite differences in the original five variables before PCA: salt type, valency, concentration, channel height, and channel material.

Figure R15: Principle component analysis of memristive loops. We reduced the dimensionality from 5 dimensions (salt type, valency, concentration, channel height and channel material) to two principle components to be able to better visualize the agglomeration of the phenomena

Question 3.3

The formation of Bjerrum pairs was also proposed in previous papers as a potential origin of memory in nanofluidic channels, but the same authors previously stated that "this process can only take place in thinner channels—those less than 2 nm in thickness—as Bjerrum pairs only exist under strong confinement". In a nutshell, the existence of Bjerrum pairs in nanoslits was not demonstrated and it is very unlikely to occur for taller channels. Therefore, the subsequent reasonings invoking this phenomenon, including the interpretation of temperature trends, are not on firm grounds.

We thank the reviewer for their insightful comment. We agree that Bjerrum pairs are theoretically expected only under strong confinement in channels thinner than 2 nm. However, in our experiments, we observed a potential “surface Wien effect” in channels up to 10 nm, which could allow Bjerrum pairs to exist in the water layers adjacent to the channel walls. The dielectric constant of the first water layers is decreased which allows this pairing near the channel walls. We have clarified in the revised manuscript (page 13) that this phenomenon remains hypothetical but is supported by our observation of a stronger Wien effect with decreasing temperature, potentially linked to the formation of Bjerrum pairs. This finding experimentally supports the 2021 theory by Robin et al. [29], which related Bjerrum pairs to temperature trends, though their presence in our system requires further investigation. We also highlighted that the surface Wien effect is conditional on specific factors, such as high salt concentration and time-varying voltage, and stressed the broader implications of our results for exploring out-of-equilibrium processes near solid-liquid interfaces.

We modified the text in page 15 in addition to what we showed in our reply to question 1, mainly related to the temperature effect as follows:

We can explain the observations with temperature variation by correlation with the mechanism behind both memristor styles. In the Wien style, Robin et al. [29] theoretically correlated the formation of

Bjerrum pairs with the reduced temperature, noting that lower temperatures promote their formation. Our experimental results confirmed this hypothesis where a larger disparity between the low and high conductive states and a stronger memristive effect were observed by decreasing the temperature. As temperature rises, the formation of Bjerrum pairs is not favourable where the thermal fluctuations becomes more favourable than the electrostatic interactions, increasing the presence of free ions and diminishing the presence of Bjerrum pairs and polyelectrolyte (Figure S36B). On the other hand, in the saturation style, temperature significantly influences ionic mobility, with higher temperatures increasing diffusion rates and expediting the formation of the depletion zone at the channel's entry, thereby enhancing loop formation (Figure S36A).

We also added figure S36 to better explain the effect of temperature

Question 3.4

The emergence of long term memory is not quantitatively compatible with the models considered in the text and in Fig. 3, see again the Science paper by the same authors.

The reviewer is right about the discrepancy between the theory time constant (in ms) and the practically observed time constants (min to hours), where the experimental time constants are several orders higher than the predicted theoretical ones, which infer the naming of long-term memory. Observing this long term memory in all of the four memristor types is quite interesting. Although we classified two types of memories in this article, long and short term, the time notion was compared to the biological systems where short term corresponds to minutes and long term to hours (rather than compared to theoretical models in which they are considered both long-term memory). Actually, in crossing (adsorption) memristor (M1 and M3), our short-term memory is several orders higher than theory diffusion timescales (L^2/D). The adsorption rate, which dominates over bulk diffusion, significantly prolongs ion residence times within the channels. This effect scales with the surface-to-bulk ratio, quantified by the Dukhin number (Du), which indicates the dominance of surface interactions in controlling the overall kinetics. In Wien memristor, the pairing-unpairing events slow down ion transport dramatically, resulting in memory retention that surpass typical diffusion-limited processes. The system's memory timescale was amplified by the balance between pairing and unpairing rates, leading to time scales orders of magnitude longer than anticipated. We added and referenced the explanation of long term memory in the revised manuscript.

The main text on page 10 is modified as follows:

For the memory effect to occur, two conditions should be met. The first one is having two distinct conductance states (for example enrichment and depletion presented in figure 3). However, this alone is not sufficient to establish the existence of a memristor as many reported RF variation experiments in literature, have not demonstrated memristive effect. The second factor is a variation in kinetics, which is essential for observing the hysteresis characteristic of a pinched loop. In the case of crossing memristors the kinetic variation is attributed to the adsorption/desorption of ions on the nanochannel walls. Robin et al.[30] discussed the case of the adsorption of cations in highly negative charge activated carbon nanochannels and its role in the observation of long-term memory. We propose a similar phenomena of adsorption/desorption to be occurring even after surface charge inversion, though involving the corresponding anions. This mechanism is likely responsible for long term memory observed in hBN nanochannels reflected by high memory time constants (See figures S8-S10).

Question 3.5

On the simulated action potential section, there are some remarks on the protocol. The classical Hodgkin-Huxley model comprises two different memristors mimicking two different channels (K and Na) and a constant current is applied. The proposed protocol, instead, implies an ad-hoc current protocol which also has a threshold. It is therefore not clear whether the resulting all-or-none behaviour follows from the properties of the memristors or from the specific protocol adopted.

We agree with the reviewer 3 (and similarly with question 4 from reviewer 1) regarding the observation that our protocol employs variable current programming rather than a constant current application as in biological nanochannels. The variation of the resulting voltage would depend on the memristive characteristic of the nanochannel and depending on the threshold we implement, the next steps in the current program will be implemented (Figure R16). i.e. If the threshold is reached, the algorithm applies a current sweep variation to a fixed value of 200 nA, regardless of the resulting potential (which is higher for more confined systems and for higher current sweeps). This process leads to a consistent potential difference across the nanochannel, even if the initial current sweep was higher (ex: 16, 24 or 32 nA). This behavior reflects the "all-or-none" law. Conversely, if the threshold is not attained, the current will reset to zero and the potential difference will dissipate.

We have removed the section related to mimicking action potentials, to avoid potential confusion for readers regarding the applied protocol. Future studies could explore the application of constant current across multiple nanochannels, offering further insights into these phenomena

Figure R16: Flowchart of the algorithm applied to mimic biological action potential. An initial current sweep is applied. The initial stimulus depolarization is simulated by a minor increase of the applied current (up to 1-32 nA), while simultaneously measuring the voltage across the nanochannel. If a predetermined threshold value (100 mV) is not reached, the current resets to zero. If the threshold is met, a higher current sweep is applied (200 nA) simulating the opening of additional voltage-gated sodium channels. The algorithm then shifts to a negative current (-15 nA) to simulate the opening of potassium channels during the repolarization and hyperpolarization steps. The current then decreases in absolute value to 0V in a lower slope reflecting the slow closure of the potassium channels.

In conclusion, the paper contains promising results but, in the current form, does not provide sufficiently substantiated insights into the physical mechanisms at the basis of the memristive behaviour.

We thank again the reviewer for his insightful and constructive comments. We have now thoroughly revised the manuscript as per the comments.

References

1. Cao, L., et al., *Concentration-gradient-dependent ion current rectification in charged conical nanopores*. *Langmuir*, 2012. **28**(4): p. 2194-2199.
2. He, Y., et al., *Tuning transport properties of nanofluidic devices with local charge inversion*. *Journal of the American Chemical Society*, 2009. **131**(14): p. 5194-5202.
3. Li, S.X., et al., *Direct observation of charge inversion in divalent nanofluidic devices*. *Nano letters*, 2015. **15**(8): p. 5046-5051.
4. Li, Y., et al., *Electrical Field Regulation of Ion Transport in Polyethylene Terephthalate Nanochannels*. *ACS Applied Materials & Interfaces*, 2019. **11**(41): p. 38055-38060.
5. Morikawa, K. and T. Tsukahara, *Shift of charge inversion point of a trivalent ion solution in a nanofluidic channel*. *Colloid and Interface Science Communications*, 2022. **50**: p. 100646.
6. Han, T., et al., *Counterion Distribution in the Stern Layer on Charged Surfaces*. *Nano Letters*, 2024. **24**(34): p. 10443-10450.
7. Van der Heyden, F.H., et al., *Charge inversion at high ionic strength studied by streaming currents*. *Physical review letters*, 2006. **96**(22): p. 224502.
8. Rezaei, M., et al., *Viscous interfacial layer formation causes electroosmotic mobility reversal in monovalent electrolytes*. *Physical Chemistry Chemical Physics*, 2018. **20**(35): p. 22517-22524.
9. López, M.L., M. Queralt-Martín, and A. Alcaraz, *Experimental demonstration of charge inversion in a protein channel in the presence of monovalent cations*. *Electrochemistry communications*, 2014. **48**: p. 32-34.
10. Calero, C., J. Faraudo, and D. Bastos-Gonzalez, *Interaction of monovalent ions with hydrophobic and hydrophilic colloids: charge inversion and ionic specificity*. *Journal of the American Chemical Society*, 2011. **133**(38): p. 15025-15035.
11. Xiong, T., et al., *Ion current rectification: from nanoscale to microscale*. *Science China Chemistry*, 2019. **62**: p. 1346-1359.
12. de Vos, W.M. and S. Lindhoud, *Overcharging and charge inversion: Finding the correct explanation (s)*. *Advances in colloid and interface science*, 2019. **274**: p. 102040.
13. Fuest, M., et al., *Cation Dependent Surface Charge Regulation in Gated Nanofluidic Devices*. *Analytical Chemistry*, 2017. **89**(3): p. 1593-1601.
14. Robin, P., et al., *Long-term memory and synapse-like dynamics in two-dimensional nanofluidic channels*. *Science*, 2023. **379**(6628): p. 161-167.
15. Ling, Y., et al., *Single-Pore Nanofluidic Logic Memristor with Reconfigurable Synaptic Functions and Designable Combinations*. *Journal of the American Chemical Society*, 2024.
16. Yossifon, G., Y.-C. Chang, and H.-C. Chang, *Rectification, Gating Voltage, and Interchannel Communication of Nanoslot Arrays due to Asymmetric Entrance Space Charge Polarization*. *Physical review letters*, 2009. **103**(15): p. 154502.
17. Cao, E., et al., *Ion Concentration Polarization Tunes Interpore Interactions and Transport Properties of Nanopore Arrays*. *Advanced Functional Materials*, 2024. **34**(11): p. 2312646.
18. Zhao, W., et al., *Evidence of Formation of Monolayer Hydrated Salts in Nanopores*. *Journal of the American Chemical Society*, 2022. **144**(41): p. 18976-18985.
19. Zhao, W., et al., *Two-dimensional monolayer salt nanostructures can spontaneously aggregate rather than dissolve in dilute aqueous solutions*. *Nature communications*, 2021. **12**(1): p. 5602.
20. Liang, X., et al., *Formation of compounds with diverse polyelectrolyte morphologies and nonlinear ion conductance in a two-dimensional nanofluidic channel*. *Chemical Science*, 2024. **15**(21): p. 8170-8180.
21. Feng, J., et al., *Observation of ionic Coulomb blockade in nanopores*. *Nature materials*, 2016. **15**(8): p. 850-855.
22. Xie, Y., et al., *Surface-charge governed ionic blockade in angstrom-scale latent-track channels*. *Nanoscale*, 2023. **15**(21): p. 9560-9566.
23. Siwy, Z.S., et al., *Negative Incremental Resistance Induced by Calcium in Asymmetric Nanopores*. *Nano Letters*, 2006. **6**(3): p. 473-477.
24. Lin, C.-Y., et al., *Electrodifusioosmosis-induced negative differential resistance in pH-regulated mesopores containing purely monovalent solutions*. *ACS applied materials & interfaces*, 2019. **12**(2): p. 3198-3204.
25. Li, D., *Electroosmotic Flow and Electrophoresis in Nanochannels*, in *Electrokinetic Microfluidics and Nanofluidics*. 2022, Springer. p. 107-147.

Reviewer 3 comments

26. Haywood, D.G., Z.D. Harms, and S.C. Jacobson, *Electroosmotic flow in nanofluidic channels*. Analytical chemistry, 2014. **86**(22): p. 11174-11180.
27. Yang, R., et al., *Negative Differential Resistance in Conical Nanopore Iontronic Memristors*. Journal of the American Chemical Society, 2024. **146**(19): p. 13183-13190.
28. Chang, H.-C. and G. Yossifon, *Understanding electrokinetics at the nanoscale: A perspective*. Biomicrofluidics, 2009. **3**(1).
29. Robin, P., N. Kavokine, and L. Bocquet, *Modeling of emergent memory and voltage spiking in ionic transport through angstrom-scale slits*. Science, 2021. **373**(6555): p. 687-691.
30. Robin, P., et al., *Long-term memory and synapse-like dynamics in two-dimensional nanofluidic channels*. Science, 2023. **379**(6628): p. 161-167.
31. Paulo, G., et al., *Hydrophobically gated memristive nanopores for neuromorphic applications*. Nature Communications, 2023. **14**(1): p. 8390.

Reply to reviewers

Programmable memristors with two-dimensional nanofluidic channels

Abdulghani Ismail^{1,2}, Gwang-Hyeon Nam^{1,2}, Aziz Lokhandwala,^{1,2} Siddhi Vinayak Pandey,^{1,2} Kalluvadi Veetil Saurav,^{2,3} Yi You,^{1,2} Hiran Jyothilal,^{1,2} Solleti Goutham^{1,2}, Ravalika Sajja^{1,2}, Ashok Keerthi^{2,3,4}, Boya Radha^{1,2,4*}

¹Department of Physics and Astronomy, School of Natural Sciences, The University of Manchester, Manchester M13 9PL, United Kingdom

²National Graphene Institute, The University of Manchester, Manchester M13 9PL, United Kingdom

³Department of Chemistry, School of Natural Sciences, The University of Manchester, Manchester M13 9PL, United Kingdom

⁴Photon Science Institute, The University of Manchester, Manchester M13 9PL, United Kingdom

* Correspondence to be addressed to: radha.boya@manchester.ac.uk

General comment

We thank the reviewers for their time, effort and thorough evaluation of our manuscript after the first round of reviewing and for their constructive feedback, which has helped refine our work.

In response, we have made **substantial revisions**. We **performed simulations** that revealed the **four memory effects**, using minimal model equations that account for ion–ion interactions, ion–surface interactions, and external polarization to explain the observed behaviors. We also **clarified the novelty** of our findings by **distinguishing our Wien-type hysteresis from previously reported phenomena**. Furthermore, we **expanded the discussion** on the mechanisms underlying the different memory effects and outlined perspectives for future in situ characterization techniques. **Additional experimental evidence** has been incorporated, **key and supplementary figures have been revised** (including the **addition of two main figures including simulations and application**), **relevant references** have been added, and minor errors have been corrected.

We are confident that these revisions comprehensively address the reviewers' concerns and significantly strengthen the manuscript. Below, we provide a detailed, point-by-point response to each comment. We highlight our responses in **blue** and note the corresponding changes to the manuscript **in highlighted yellow**, for ease of reference.

Reviewer #1:

The revised manuscript entitled Programmable memristors with two-dimensional nanofluidic channels described a 2D nanofluidic memristor showing all 4 kinds of conductivity switch kinetics in different electrolyte environment. Undoubtedly, memristors with this programming ability was not reported in both nanofluidic systems and solid memristor devices. Generally, the manuscript was properly revised and all the raised problems were solved in the revised manuscript. This revised manuscript deserve publication in Nat. Commun., while I still have several minor questions which I hope can be taken into discussion.

We sincerely appreciate your positive evaluation of our revised manuscript. We are glad to hear that you found our revisions satisfactory, and that the manuscript is now **suitable for publication in Nature Communications**. We also appreciate your additional minor questions and welcome the opportunity to further clarify and discuss any remaining points. Please find our detailed responses below, addressing each of your questions:

Question 1.1

It seems that when there is almost no charge effect, Wien effect governed the observation while when charge effect governed the channel, the occurrence of M1, M2 or M3 is related to the surface-ion interaction intensity. For monovalent cations, the surface-ion interaction is limited, thus in most cases M4 was observed; And for multivalent cations, stronger interaction contributed to the concentration-dependent switch among M1, M2 and M3, right? Personally, I recommend the authors to give a brief conclusion on what interaction controlled the changes of memristor types among M1, M2, M3 and M4. Again, in what cases, Wien effect could govern the whole system? It seems that there should be both strong charge inversion (surface-ion interaction) and strong ion-ion interaction in concentrated electrolyte. What parameter decided which one is the major effect?

Thank you for your insightful question and recommendation. You are correct in noting that the occurrence of different memristor types (M1, M2, M3, and M4) is largely governed by the interaction between surface charges and ions in the nanochannel.

As you mentioned M4 occurs mainly for monovalent ions at high concentrations as they show least interaction with surface charges (thus no crossing at high concentration). Bi- and tri-valent ions shows crossing due to surface-charge interaction. Additionally, the interaction between ions themselves plays an important role in determining the resulting memory; The interplay between these interactions (surface charge-ion and ion-ion) is what is determining the final output (please see new simulations in SI which are also added to this response).

At low concentrations, the surface charge has more effect, which result in crossing for monovalent (also the M4 effect decreases due to less ions). Interestingly, by decreasing the pH (less surface charge) at low concentrations there is a shift from crossing 2 to Wien confirming the M3 effect is due to surface charge (Figure SI 15B).

When surface charge effects dominate, the occurrence of M1, M2, or M3 depends on the surface-ion interaction intensity:

- 1) Monovalent cations exhibit weaker interactions than multivalent with the nanochannel walls, leading to predominant M4 behaviour at high concentrations. However, at lower concentrations, surface effects can become more significant, leading to crossing-type behaviour (M1 or M3).

Reviewer 1 comments

- 2) Multivalent cations interact more strongly with the surface, leading to concentration-dependent transitions among M1, M2, and M3. In particular:
- M1 (Crossing 1) and M3 (Crossing 2) are linked to surface charge inversion effects at different concentration ranges, where the main charge carrier (anion or cation) switches. The observation of rectification due to asymmetric conditions and of different kinetics governed by variable adsorption/desorption dynamics is at the basis of their occurrence.
 - M2 (Saturation) is observed in cases where ion concentration polarization plays a dominant role, particularly with multivalent cations that have polymeric forms (e.g., Al^{3+} or Mn^{2+}), which promote surface charge inversion and depletion-layer effects at the channel entrance at high voltages.

All the four effects necessitate extreme confinement; otherwise, the system will not show any memory.

We **added a brief summary** in the revised manuscript to highlight the governing interactions that dictate the transition between these memristor types as well as a scheme to facilitate the understanding. Revised in pages 14 to 17 of main manuscript:

To summarize the various effects that determine the memory type - M4 (Wien effect) memristor is observed when the ions interact with themselves more than with the surface charge, allowing the applied electric field to dictate ion dynamics, particularly in high monovalent electrolyte concentrations where the channel behaves as a near-neutral system. This fact is consolidated by the shifting of the governing effect from crossing 2 to Wien when the surface charge is reduced by decreasing pH at low electrolyte concentrations (supporting information Figure S15). When surface charge effects dominate, the occurrence of M1, M2, or M3 depends on the surface-ion interaction intensity:

- Monovalent cations exhibit weaker interactions than multivalent cations with the nanochannel walls, leading to predominant M4 behavior at high concentrations. However, at lower concentrations, surface effects can become more significant, leading to crossing-type behavior (M1 or M3).
- Multivalent cations interact more strongly than monovalent cations with the surface charges, leading to concentration-dependent transitions among M1, M2, and M3. In particular:
 - M3 (Crossing 2) transitions to M1 (Crossing 1) due to SCI, where the main charge carrier (anion or cation) switches. The observation of rectification due to asymmetric conditions and kinetics governed by variable adsorption/desorption dynamics is at the basis of their occurrence.
 - M2 (Saturation) is observed in cases where ion concentration polarization plays a dominant role, particularly with multivalent cations that have polymeric forms (Al^{3+} or Mn^{2+}), which promote surface charge inversion and depletion-layer effects at the channel entrance at high voltages.

All four effects necessitate extreme confinement; otherwise, the system will not show any memory. This systematic interplay between confinement, electrolyte concentration, and surface charge effects highlights the tunability of the nanofluidic memristor, and controlled variations in pH, electrolyte type, and confinement conditions can enable transitions between different memristor behaviors.

Figure R1 (shown as table of contents scheme): Schematic representation of the main factors governing the occurrence of M1-M4 memory styles, including ion-surface charge interactions, ion-ion interactions, ionic flux, and nanochannel confinement.

Furthermore as indicated in the reply to reviewer 3, we **performed simulations** to correlate the effect of interaction of various ions between themselves, with the surface charge and with the external depletion region. By varying the parameters of adsorption/desorption and ion interaction, we were able to **observe the memory shift that we obtained experimentally**. The discussion is now enriched in the main and in the SI concerning this model. We also added a main figure obtained by simulations.

We added the following discussion and figure to the revised main article, pages 15 to 17:

Building on the qualitative insights described above, we propose a minimal theoretical model that quantitatively captures the memristive effects for the four observed loop types (M1, M2, M3, and M4), as detailed in the SI. This model is an extension of the minimal model previously developed by Robin et al.⁷⁰, now including additional parameters explicitly accounting for ion-ion and ion-surface interactions, and reservoir depletion effects. By tuning these parameters within the same framework, we successfully simulated all four experimentally observed memristive loops that coexist in the same device but to different extents (Figure 5). This unified theoretical approach helps in the quantitative explanation of the diverse memristive behaviors observed.

From the preceding discussions, the memory in 2D nano/angstrom channels is governed by ion-ion interactions, ion-channel wall interactions, and channels' entrance depletion. Each of these interactions has a competing nature and takes place at varying but comparable timescales. Adsorption (α_A, β_B) and desorption (l_A, l_B) rates of anions and cations to the wall charges regulate the ion-channel wall interactions (Figure 5). The interaction strength parameter, $\delta_0[f]$, governs the ion-ion interactions, where the Wien effects cause a voltage dependence (Figure 5A, E). The external coupling γ governs the depletion of ions occurring exterior to the channel explaining the saturation memristor type (M2) (Figure 5A,C). Further discussions about each term are in Supplementary section 2.

Figure 5: Simulation of four memristive loops using minimal model. A) Schematic illustrating the various parameters used in the differential equations governing ion interactions inside the nanofluidic channel (See SI section 2). These include the adsorption (α_A, β_B) and desorption (l_A, l_B) rates of anions and cations to the wall charges, the ion-ion interaction term ($\delta_o[f]$) and channel entrance depletion parameter (γ). Under an external forcing function $f(t)$, cations and anions migrate in opposite directions from two reservoirs with different concentrations (c_r and c_l), reflecting the asymmetric entrance of the nanochannel. (B,C,D,E) show the corresponding I–V and G–V curves, respectively, obtained using the equations in SI section 2 by varying the interaction parameters. This results in four distinct memory effects (M1–M4). The specific parameters used for each plot are indicated within the figure panels.

Question 1.2

One interesting thing is that the device showed LTP (M3), LTD(M1), STP(M4) and STD(M2) in the same voltage in different electrolyte, why the conductivity switch can be (cannot be) hold without bias voltages should be briefly discussed.

We appreciate the reviewer's insightful observation. The ability of the device to exhibit long term memory (LTP, LTD) or short-term memory (STP, STD) at the same applied voltage but with different electrolytes highlights the tunability of nanofluidic memristors based on electrolyte composition and ion interactions.

The key factor determining whether conductivity switching can be retained without bias voltage is the underlying memory mechanism associated with each memristor type:

- Long-term memory (M3) arises from charge adsorption/desorption effects on the nanochannel walls, which persist even after the removal of the applied voltage. This is particularly evident in crossing-type (M3) memristors, where the ion adsorption is in a stable state, maintaining memory even after bias removal. Ion desorption is achieved by applying opposite polarity rather than simple relaxation at zero-volt.
- Short-term memory (M2, M4) is governed by either ion depletion at the channel's entry (M2, saturation-type) or polyelectrolyte formation (M4, Wien effect), both of which rely on ion redistribution that quickly relaxes once the voltage is removed. Since these effects are transient and linked to ionic mobility rather than adsorption, memory is lost once the bias is turned off.

Thus, the distinction between surface charge-driven (long-term) and ion dynamics-driven (short-term) mechanisms explains why some memory states persist after removing the bias, while others decay over time.

We added figure 6 to explain the difference behaviour of short/long term depression/potential that are observed across the different memristors, along with explanation of the correlation with the mechanism and biological phenomena. Pages 17 to 20 in main manuscript revised as below.

Biological synapses and neurons are natural counterparts to nanofluidic memristors, facilitating memory processing in the brain. We attempt to emulate biological processes especially at the level of synaptic conduction using our 2D nanochannels, such that an increase of nanochannel conductance resembles an increase in the synaptic weight. Various memristive behaviors observed in our artificial 2D nanochannels draw parallels with synaptic phenomena observed in biological neural systems. We report volatile (short-term) and non-volatile (long-term) memories (Figure 6) as well as potentiation and depression behaviors.

Short-term potentiation (STP) is a form of volatile memory that is characterized by an initial increase in synaptic conductance upon stimulation followed by a spontaneous decay once the stimulus is removed. We observed two cases of STP:

- Case 1 of low read voltage, high write voltage (Figure 6A) This behavior is reminiscent of excitatory postsynaptic currents (EPSCs) in biological neurons, that reflect synaptic activation and contribute to temporal signal processing and synaptic filtering. The volatile nature of this behavior is inferred from the return to the initial conductance level after relaxation at zero volts (Figure 6A). This response is observed in M4, where the underlying mechanism involves the formation of a polyelectrolyte upon the application of a writing stimulus, which subsequently dissipates when the device is relaxed at zero volts.

- Case 2: High read voltage - relaxation at zero volts (Figure 6B). This type of STP is observed in the M2 memristor, which remains under a high reading potential and thus functions as both a reading and writing potential. Once the device is relaxed at zero volts for a certain duration and subsequently re-exposed to the high reading potential, the conductance of the nanochannel temporarily increases before returning to its pre-relaxation value. This could be explained by the disruption (at zero volt) of the depletion zone at the channel entrance, which forms under high voltage over an extended period. During relaxation, the ion concentration recovers to pre-excitation levels, leading to a temporary increase in conductance. However, upon reapplying the potential, the depletion zone at the channel entrance reforms, resulting in a decrease in conductance.

Short-term depression (STD) is another form of volatile memory that is characterized by an initial decrease in conductance upon stimulation followed by a spontaneous increase back to initial values once the stimulus is removed. Here also, we observe two cases:

- Case 1: Low read voltage - high write voltage (Figure 6C). This behavior is reminiscent of inhibitory postsynaptic currents (IPSCs) in biological neurons, essential for neural network modulation and preventing overstimulation. This behavior is observed in M1, where the underlying mechanism is likely related to the channel depletion and desorption of ions upon the application of a positive polarity, leading to subsequent decrease in conductance. Most of this type of memory disappears after a few minutes of stopping the excitation; however, a slight long-term memory effect is noticeable, where the stable conductance after reading remains slightly lower than its pre-reading value. This phenomenon follows the same trend as crossing 2 (Figure 6G), where both short- and long-term memory exist, but in this case, the long-term effect is negligible at the applied durations, likely due to the high electrolyte concentration, which facilitates ion diffusion into the nanochannel and erases the long-term impact of the applied stimulus.
- Case 2: of high read voltage-relaxation at zero volt (Figure 6D). The results are inverse of Figure 6B. The inhibitory effect seen in the Wien memristor echoes the Na⁺ current behavior after depolarization (to 0 mV from resting potential -75 mV) of voltage-gated sodium channels in axon's initial segment⁷¹. The current variation after turning off the power source is linearly proportional to the relaxation time on a logarithmic scale, suggesting an exponential decay, the longer the waiting time is, the more the current is inhibited (i.e., more of the polyelectrolyte is dissociated) (inset of figure 6D).

Long-term potentiation (LTP), a cornerstone of learning and memory in biological systems, is effectively represented in panels 6E and F through non-volatile memory behaviors. These curves demonstrate sustained increases in conductance persisting from minutes (Figure 6E, M3) to days (Figure 6F, M3, 74 hours). Here, applying an erasing negative polarity voltage resulted in the loss of stored memory and a return to the low-conductance state (Figure 6F). This prolonged retention of elevated conductance levels even when the device is disconnected, closely aligns with biological synaptic strengthening, which underlies memory formation and long-term learning. These stable states highlight the potential of our 2D nanochannels to emulate synaptic plasticity and enable in-memory information storage. Figure 6G shows both short-term and long-term potentiation in M3 memristor, governed by writing pulse duration. Short pulses produce transient conductance (short-term memory), while longer pulses induce a short-term part that disappears after short duration in the range of tenth of seconds as well as lasting conductance changes (long-term memory). This dual behavior is essential for neuromorphic systems combining working memory and stable learning.

Reviewer 1 comments

The ability of the device to exhibit different memory behaviors under the same applied voltage but with different electrolytes highlights the crucial role of ion dynamics and surface interactions in defining memory retention. Long-term memory (M3) arises from charge adsorption/desorption effects on the nanochannel walls, resulting in stable states that persist even after the removal of the applied voltage, however, it can be removed by applying opposite polarity potential. In contrast, short-term memory (M2, M4) is governed by ion depletion at the channel entrance or polyelectrolyte formation, both of which involve dynamic ion redistribution that quickly relaxes once the voltage is removed. This distinction between surface charge-driven (long-term) and ion dynamics-driven (short-term) mechanisms explains why some memory states are retained without bias, while others decay over time, offering tunable control over memory timescales in nanofluidic memristors. In the case of Wien memristor, erasing and restarting the system is thus done by simple relaxation at zero volt, while in the case of crossing memristor it is done by application of voltage of opposite polarity (Figure S30 and S32). Both memristors response depends on the number of pulses, duration of pulsing (or relaxation) and on the history of voltage application (Figure S30-33 and Figure 6G), which implies a significant potential to contribute to various elements in the realm of neuromorphic computing.

Volatile memory

Non-volatile memory

Mixed

Figure 6: Volatile and non-volatile memory. A) Excitatory action in a MoS₂ device (h = 0.7 nm, 1 M KCl) demonstrating a Wien memristor (M4) loop. The retained memory after applying a positive potential pulse of 1 V is partially erased and then completely erased after relaxation at 0 V for 1 min and 5 min, respectively, transitioning the system from LRS to HRS. The reading voltage is 500 mV. B) Excitatory action in a hBN device (h = 0.7 nm, AlCl₃ 10 mM) demonstrating saturation-type memory (M2) after a 20-minute relaxation at 0 V, read at 1 V. C) Inhibitory action in an hBN device (h = 2 nm, AlCl₃ 1 M) demonstrating a crossing 1 memory (M1) after a writing pulse of 0.8 V, read at 50 mV. Residual long-term memory persists, as the conductance does not return to its original value. D) Inhibitory action in an hBN device (h = 1.4 nm, 3 M KCl) showing Wien-

type memory. The device is relaxed at 0 V and read at 1 V. The current magnitude variation relative to its equilibrium value is plotted against relaxation time in semi-log coordinates (inset). This case is the opposite of B. E) Non-volatile memory in MoS₂ nanochannels (h = 0.7 nm, 0.01 M KCl), showing a crossing 2 memory (M3). The retained memory after a positive pulse of 0.12 V is not erased after relaxation at 0 V for 5 min. The reading voltage is 30 mV. F) Non-volatile memory in MoS₂ nanochannels (h = 0.7 nm, 0.1 M KCl), showing a crossing 2 type memory (M3). The retained memory after a positive pulse of 0.2 V is not erased after relaxation at 0 V for ~ 3 days. The reading and erasing voltages are -20 mV and -200 mV respectively. G) Coexistence of short- and long-term memory in MoS₂ nanochannels (h = 0.7 nm, 0.01 M KCl), showing M3-type behavior. Varying pulse durations lead to both short (<20 s) and long-term memories are observed. Reading at -0.03 V preserves state; writing is done with +0.1 V pulses.

Question 1.3

Page 11 Line 17 It seems that there is a typo error that the authors is giving description to Fig. 4C instead of 3D right ? :

We appreciate the reviewer's attention to detail. Indeed, there was a typographical error in the figure reference. The correct reference should be Figure 3D instead of Figure 4C. We have now corrected this in the revised manuscript.

Reviewer #2:

In the revised manuscript, the authors have now clarified the asymmetric geometry of their nanochannel and do agree that the rectification phenomena they see are due to asymmetric external polarization at the two asymmetric entrances of their channel. This topic was first studied by Yossifon et al in their 2009 PRL paper (ref 56) and then by many others. Yossifon et al also used straight channels like the authors, not conic nanopores. The authors have made smaller nanochannels to allow for lower voltage and higher ionic strength. They have also used results from earlier reports to interpret their data. Nevertheless, it is not that novel. The other claimed novelties of the paper are memory effect due to salt condensation/charge inversion, particularly monovalent salt concentration like K⁺, and hysteresis and memory due to Wien water dissociation. The former charge inversion phenomenon has also been reported by many researchers, including the authors. The small nanoachannels in the present study have, nevertheless, allowed for a larger class of charge inversion phenomena and the authors do offer a more comprehensive classification of the resulting different memory effects. Their most novel results, however, are the M4 hysteresis by the Wien effect. The only known literature report of Wien hysteresis I am aware of are by Cheng and Chang (Biomicrofluidics, 5, 046502(2011); LabChip, 14, 979(2014)) with a theoretical explanation of the hysteresis due to concentration front propagations by Conroy et al (PRE, 86, 056104(2012)). The authors have now focused their manuscript on this less-studied hysteresis effect but they need to compare their results to these earlier studies. If they can add these further revisions on the M4 wien hysteresis, I believe the paper would then be acceptable for publication.

We thank the reviewer for their positive consideration of our work for publication following revisions, and confirming the novelty of our work, particularly in observing new memory phenomena in highly confined systems, including monovalent charge inversion and memory effects.

We also appreciate the reviewer highlighting the importance of Wien-type hysteresis (M4). We acknowledge the relevant studies by Cheng, Chang, and Conroy, and have now **incorporated a comparison with these prior reports in the revised manuscript**. In Conroy et al. article, at the junction of a cation and anion exchange membrane, upon applying voltage bias, the transient depletion of mobile ions at the membrane junction was shown to create time-dependent hysteresis in the I-V characteristics. Further, at high voltages (10^7 V/cm), field-enhanced water dissociation phenomenon led to increased currents, with propagation speeds of the proton and hydroxyl ion fronts controlled by the ion current. These effects were manifested for electrokinetic control of fluid pH and generation of stable pH gradients (Biomicrofluidics, 5, 046502(2011)). In our current study, we have dissociation and association of ions primarily due to electric fields ($\sim 0.5\text{V}/5\times 10^{-4}\text{cm} \approx 10^3$ V/cm) in nanoconfinement, and these association/dissociation kinetics led to memory effects, short term potentiation (Supplementary Figure S30) along with reproducible conductance switching as shown by the endurance graphs (Supplementary Figure S29). The electric field is lower and insufficient to dissociate covalent water bond (bond energy ~ 498 kJ/mol) while it will permit the dissociation of ionic Bjerrum pairs which have at least one order of lower bond energy^{3,4}. Furthermore, the hysteresis observed in our system is very robust and reproducible (Figure SI 28-29).

Additionally, to rule out the possibility that our observed Wien effect results from water splitting, we now present I–V curves for pure water (without any added salt) (Figure R2). These results demonstrate that in the absence of salt, the I-V curve remains purely capacitive, with no Wien memory effect or hydrolysis observed. Our confined nanochannels **permit the existence of these Bjerrum pairs** (due to lowered water dielectric constant in confinement), which have lower energy to break compared to water covalent bond. Our nanochannels system allows us to remain within the

Reviewer 3 comments

stable water activity range (much lower than 1.23 V vs. SHE, where water hydrolysis occurs) while still achieving strong memory effects. This further highlights the significance of the one to two orders of magnitude voltage gain that we achieve with our nanofluidic system, making it a more energy-efficient platform for exploring memristive behaviors.

Figure R2: IV characteristic of water inside 0.7 nm MoS2 nanochannel showing capacitive behaviour.

We included the comment of the reviewer and compare our polyelectrolyte Wien to the water Wien reported in the Cheng, Chang and Conroy articles by modifying the main text page 12 as follows:

Hysteresis in I-V curves induced by Wien effect and controlled by ion concentration propagation was reported previously⁶²⁻⁶⁴. In this case, the hysteresis occurs due to ion depletion at the bipolar membrane junction leading to extremely high electric fields ($\sim 10^7$ V/cm) which emerge after salt depletion under modest voltage bias⁶⁰⁻⁶². In contrast, our system operates under electric fields $\sim 10^3$ V/cm, and within the stable water activity range. Additionally, control experiments using pure water (without added salt) resulted in pure capacitive IV curve without any memory or hydrolysis effects. This confirm our interpretation that the reason behind the current Wien effect under high confinement is polyelectrolyte Wien effect driven by ion dissociation rather than water splitting.

Reviewer #3:

Question 3.1

Unfortunately, the feedback on this work cannot change from the previous submission: it contains promising results but the explanations remain qualitative. No simulations, theory, or models were added. Since the different reported regimes rely on a subtle balance between different phenomena, this is a serious drawback which prevents a broad and reproducible impact of these insights.

We fully appreciate the concerns regarding the absence of a quantitative modeling and simulations in our previous reply, to complement our experimental results. To address the concerns of the reviewer, we have now introduced a **theoretical framework** accompanied by **numerical simulations**,

Reviewer 3 comments

that explain and simulate the four memristive loops, and included the description in the revised manuscript (Figure 6) and supplementary information

The theoretical model quantitatively captures all four experimentally observed memristive behaviors (M1, M2, M3, and M4). This model is governed by coupled ordinary differential equations that precisely describe ionic transport dynamics, ion-wall adsorption/desorption kinetics, ion-ion interactions (including ion pairing), and reservoir depletion effects. Each mechanism's contribution could now be probed mathematically by varying the interaction parameters, clarifying the subtle interplay responsible for transitioning between the distinct memory regimes.

We performed numerical simulations of the developed model, demonstrating parameter regimes where each of the four memristive loops emerges. The **simulated loops closely match experimental observations (Figures 5 and 1 in main, and section 2 in SI)**, verifying the quantitative explanation of the proposed model. Parameters such as **adsorption/desorption rates, ion-pair interaction strength, and reservoir depletion factors** have been systematically varied, elucidating their quantitative influence on the channel's memristive behavior.

We added the following discussion paragraph to the revised main article, pages 15 to 18:

Building on the qualitative insights described above, we propose a minimal theoretical model that quantitatively captures the memristive effects for the four observed loop types (M1, M2, M3, and M4), as detailed in the SI. This model is an extension of the minimal model previously developed by Robin et al.⁷⁰, now including additional parameters explicitly accounting for ion-ion and ion-surface interactions, and reservoir depletion effects. By tuning these parameters within the same framework, we successfully simulated all four experimentally observed memristive loops that coexist in the same device but to different extents (Figure 5). This unified theoretical approach helps in the quantitative explanation of the diverse memristive behaviors observed.

From the preceding discussions, the memory in 2D nano/angstrom channels is governed by ion-ion interactions, ion-channel wall interactions, and channels' entrance depletion. Each of these interactions has a competing nature and takes place at varying but comparable timescales. Adsorption (α_A, β_B) and desorption (l_A, l_B) rates of anions and cations to the wall charges regulate the ion-channel wall interactions (Figure 5). The interaction strength parameter, $\delta_0[f]$, governs the ion-ion interactions, where the Wien effects cause a voltage dependence (Figure 5A, E). The external coupling γ governs the depletion of ions occurring exterior to the channel explaining the saturation memristor type (M2) (Figure 5A,C). Further discussions about each term are in Supplementary section 2.

Figure 5: Simulation of four memristive loops using minimal model. A) Schematic illustrating the various parameters used in the differential equations governing ion interactions inside the nanofluidic channel (See SI section 2). These include the adsorption (α_A, β_B) and desorption (l_A, l_B) rates of anions and cations to the wall charges, the ion-ion interaction term ($\delta_o[f]$) and channel entrance depletion parameter (γ). Under an external forcing function $f(t)$, cations and anions migrate in opposite directions from two reservoirs with different concentrations (c_r and c_l), reflecting the asymmetric entrance of the nanochannel. (B,C,D,E) show the corresponding I–V and G–V curves, respectively, obtained using the equations in SI section 2 by varying the interaction parameters. This results in four distinct memory effects (M1–M4). The specific parameters used for each plot are indicated within the figure panels.

Supporting information revision (Pages 4 to 7)

Section 2: Minimal model simulations

1. Theoretical Model of Nanofluidic Memristors: A Unified Framework for four Distinct Memristive Behaviors

Memristive behaviors observed in 2D nanochannels result from a combination of ion–ion interactions, ion–channel wall interactions, and external polarization resulting in channel entrance depletion. We build upon a minimal model to describe four experimentally observed memristive loops (M1-crossing 1), M2-saturation, M3-crossing 2, and M4-Wien), through a set of dynamical equations. We extend and unify the minimal model originally proposed in a previous article[3]. Our

extended model incorporates coupled ion transport, adsorption–desorption kinetics, ionic pairing, and voltage-driven depletion from reservoirs.

2. Mathematical Framework

In the simplest picture, the memristive response of a nanochannel can be described by an internal state variable, representing the fraction of ions actively contributing to conduction. Its time evolution is governed by adsorption-desorption dynamics at the channel walls and/or by the ionic association-dissociation within the channel. If the selectivity of these channels has to be taken into account, this can be two (or more) internal state variables representing the fraction of ions of a given polarity. The contribution of association/dissociation term can be explicit to n or can be made implicit into the F_{ads}/F_{des} . This can be expressed as follows.

$$\frac{dn}{dt} = F_{ads}(n, \sigma, V) - F_{des}(n, \sigma, V) \pm F_{ionic}(n, V) \quad (\text{eq 1})$$

where $\sigma(t)$ denotes the fraction of ions adsorbed on channel walls, F_{ionic} represents the ion-ion interaction term which can depend on voltage $V(t)$ in case of Onsager's second Wien effect, or fractional Wien effect in nanopores, or it could be voltage independent, if ions of different polarity are obeying predator/prey dynamics at low concentrations. The ionic current at each instant depends on the instantaneous internal state as follows:

$$I(t) = G_{int}(n, V) V(t) \quad (\text{eq 2})$$

The key to obtain different memristive loops lies in how F_{ads} , F_{des} , F_{ionic} and G_{int} respond differently to changes in ionic concentration, association/dissociation dynamics, surface charge, and external voltage polarity and magnitude. Considering these, below we extend our previous minimal model [3] to obtain all 4 memristive loops. Since, the higher valency electrolytes might form polymeric and oligomeric complexes, the discussion will be made for a monovalent electrolyte.

Let c_A and c_B denote the spatial average of concentrations of A^+ and B^- ions in the nanochannel. Similarly, σ_A and σ_B represent the concentrations of these ions adsorbed on the channel walls. The nanochannel is connected to asymmetric left and right reservoirs with ion concentrations c_l and c_r , and an external voltage drives ionic motion, modeled via a normalized forcing function $f(t)$. We define, $t = t_{actual} * D/L^2$, where L is the length of channel, typically, $\sim 5 \mu m$, t_{actual} in seconds and D is the diffusion coefficient. The factor D/L^2 must be considered during memory window calculation for individual switching types [3]. For sake of simplicity, we will work with dimensionless parameters throughout this discussion. Adapting the kinetic approach, the concentration of A^+ and B^- ions within the channel is governed by the following transport equations.

- Transport equations:

$$\dot{c}_A = -c_A + 0.5 * (c_l + c_r) + 0.5 * (c_l - c_r) * f(t) - (\dot{\sigma}_A) + \delta_0[f]c_Ac_B \quad (\text{eq 3})$$

$$\dot{c}_B = -c_B + 0.5 * (c_l + c_r) + 0.5 * (c_l - c_r) * f(t) - (\dot{\sigma}_B) - \delta_0[f]c_Ac_B \quad (\text{eq 4})$$

$$\delta_0[f] = \frac{\delta}{1 + \exp(\phi|f|)} \quad (\text{eq 5})$$

These equations describe:

- Natural decay to equilibrium: First term in eq (3) and eq (4) indicates the diffusive decay, and second term indicates value at equilibrium.
- Ion input from reservoirs modulated by $f(t)$: Third term in eq (3) and eq (4) indicates the changes in concentration of ionic species with forcing $f(t)$

iii) Losses due to adsorption onto walls: This is described by the fourth term in eq (3) and eq (4) which is governed by eq (6) and eq (7).

iv) Ion-Ion interaction: This is described by the fifth term in eq (3) and eq (4).

- Adsorption/Desorption dynamics:

$$\dot{\sigma}_A = \alpha_A c_A - \beta_B c_B - l_A \sigma_A \quad (\text{eq 6})$$

$$\dot{\sigma}_B = -\beta_B c_B - \alpha_A c_A - l_B \sigma_B \quad (\text{eq 7})$$

where α_A, β_B are adsorption rates, and l_A, l_B are desorption rates.

- Ionic interaction:

The ionic interaction term is considered here, by assuming that ionic interactions obey predator-prey dynamics within the channel. Since the M4 switching in the fluidic channels is associated with Wien effect, it is reasonable to assume that the strength of this term depends on applied voltage. The voltage or here the forcing dependence of this interaction term is defined as δ_0 in equation 5.

The channel conductance G_{int} is given by:

$$G_{int} = c_A e \mu_A + c_B e \mu_B \quad (\text{eq 8})$$

$$\text{Advection Flux} = G_{int} * f(t) \quad (\text{eq 9})$$

To incorporate, the depletion (M2 loop) in this model, we introduce a depletion factor $\Theta(f)$ which depends on $\gamma \in [0,1]$ defined as follows.

$$\Theta(f) = \exp(-\gamma \Omega(f)) \quad (\text{eq 10})$$

$\gamma = 0$ when channel is fully depleted, and $\gamma = 1$ when channel depletion is completely absent. Any value between zero and one, of this factor will indicate a mixed state preset within the channel. This factor enters the above equation as follows.

$$\dot{c}_A = -c_A + \Theta(f, t) \cdot [0.5 * (c_l + c_r) + 0.5 * (c_l - c_r) * f(t) - (\dot{\sigma}_A) + \delta_0 [f] c_A c_B] \quad (\text{eq 11})$$

$$\dot{c}_B = -c_B + \Theta(f, t) \cdot [0.5 * (c_l + c_r) + 0.5 * (c_l - c_r) * f(t) - (\dot{\sigma}_B) - \delta_0 [f] c_A c_B] \quad (\text{eq 12})$$

Here, we assume, $\Omega(f) = f^2$, yields:

$$\Theta(f) = e^{-\gamma f^2} \quad (\text{eq 13})$$

The modified advection flux is given as follows:

$$\text{Advection flux} = G(\Theta(f)) \cdot f(t) \quad (\text{eq 14})$$

where, G is modified conductance defined from eq(11, 12) in eq(8).

Heuristic Interpretation

- Without forcing, in absence of any interaction terms: $c_A = c_B = \frac{c_l + c_r}{2}$ in the channel, and $c_A = c_B = c_l/2$ in the left reservoir and $c_A = c_B = c_r/2$ in the right reservoir.
- With forcing, without interaction terms: Asymmetric ionic entry leads to rectification depending on $f(t)$, but cannot explain the memory in the desired memory window. The concentration (c_A, c_B) is modified as follows.

$$\text{- Left reservoir } \left(\frac{c_l}{2}(1 - f(t)), \frac{c_l}{2}(1 + f(t)) \right)$$

- Right reservoir $\left(\frac{c_r}{2}(1 + f(t)), \frac{c_r}{2}(1 - f(t))\right)$
- nanochannel $\left(\frac{c_l+c_r}{2} + \left(\frac{c_l-c_r}{2}\right)f(t), \frac{c_l+c_r}{2} - \left(\frac{c_l-c_r}{2}\right)f(t)\right)$

- Adsorption/desorption dynamics of A^+ and B^- ions in equation 6, 7 can be understood as follows:

- 1st term: Higher the concentration of A^+ ion within the channel, more chances are for A^+ ion to be adsorbed onto the surface of channel.
- 2nd term: Higher the concentration B^- ions, lesser effective it will be for A^+ ions to get adsorbed.
- 3rd term: The adsorbed ions within the channel will desorb with rate l_A/l_B .

3. Memory Loop Behaviors and Conditions

Each of the four observed memory types corresponds to specific parameter regimes within this unified model.

- M1 (Crossing 1):

- Dominated by strong A^+ adsorption/desorption.
- $\alpha_A, l_A \gg \delta; \beta_B, l_B \ll \delta; \alpha_A > \delta > \beta_B$.
- B^- adsorption minimal.
- Results in bipolar hysteresis with crossing direction reversed compared to M3.
- $\gamma=0$.
- Forcing frequency used: 60 mHz.
- Asymmetry: $c_l = 10, c_r = 100$.

- M3 (Crossing 2):

- Similar to M1 but with inverted adsorption rate hierarchy.
- $\alpha_A, l_B \ll \delta; \beta_B, l_A \gg \delta; \beta_B > \delta > \alpha_A$.
- Rectification favors the opposite voltage polarity.
- $\gamma=0$.
- Forcing frequency used: 75 mHz.
- Asymmetry: $c_l = 10, c_r = 100$.

- M4 (Wien-type):

- Dominated by ion-ion interaction (Bjerrum pairing).
- $\alpha_A \sim \delta; l_A, l_B, \beta_B < \delta; l_B < \beta_B, l_A$.
- Yields unipolar nonlinear hysteresis symmetric in voltage.
- $\gamma=0$
- Forcing frequency used: 35 mHz.

- M2 (Saturation type):

Reviewer 3 comments

- Occurs when ion supply from the external reservoir to the channel becomes limiting at high voltage with selectivity of the nanochannel akin to conditions of M1.
- $\gamma=1$, reservoir depletion
- Captures saturation and possible negative differential resistance.
- Forcing frequency used: 0.1 Hz.

4. Experimental Parameters and Loop Simulation

Although, clear switching trends are presented with zero-point initial conditions for $c_A, c_B, \sigma_A, \sigma_B$, i.e. $c_A[t = 0] = c_B[t = 0] = \sigma_A[t = 0] = \sigma_B[t = 0] = 0$, appropriate values of them were chosen (available in the code) to have zero crossing condition intact. In Figure 5, a few points were removed for values of f in range $0 \rightarrow 0.12$, to enable comparison among different switching mechanisms. This also distinguishes the presented system from an ideal memristor, as for an ideal memristor, irrespective of value of state variable at any times, at zero voltage current is zero. The observed loop shapes arise from solving the coupled ordinary differential equations for $c_A, c_B, \sigma_A, \sigma_B$ under time-dependent forcing $f(t)$. The four models (M1- M4) show distinct loop shapes in both advection flux versus f and conductance G versus f , in line with experimental observations.

This unified model captures the diverse spectrum of nanofluidic memristive responses through set of coupled equations. The memristive loop type depends critically on the relative magnitude of adsorption, desorption, and ion-pairing rates. The minimal model, when extended with voltage-induced depletion effects, also allows inclusion of saturation (M2). Together, these mechanisms explain the four principal types of memristive loops and provide the predictive power for designing programmable ionic devices using 2D nanofluidic channels.

The shortcomings of this model is with the assumption that only pairwise ionic interactions are dominant which is inspired from the Bjerrum hypothesis and predator-prey model. Together with this, the model is presented for the monovalent electrolytes, and might not be straightforward to extend for multivalent electrolytes. Further development of this model could include multiple ion-ion interactions. Additionally, the functional form introduced in eq (5), eq(10) and eq(13) is arbitrary and is based on the rationale that at higher voltages the effect of depletion of channels (or crowding of ions near the entrance) is less prominent, which could be improved in future studies.

Question 3.2

Additionally, the explanation of loop type M4 based on Bjerrum pairs/polyelectrolytes is purely hypothetical as acknowledged by the authors; the statement that "the dielectric constant of the first water layers is decreased which allows this pairing near the channel walls" in the taller channels is also potentially interesting but unsubstantiated.

It is true that simulations were at the basis of the Wien effect mechanism in nanochannels published by Robin et al (Science 2021)⁹; in our previous article (Science 2023)¹ we had experimental observation of memory loops in MoS₂ nanochannels which were correlated with the simulated loops.

Concerning the explanation of existence of Wien effect in channels up to 10 nm, we added further references that support the decrease of the dielectric constant of water in the nanochannel and mainly in the layers near the wall ^{7,10,11} and revised as below in page 13 of main manuscript

This discrepancy may be due to the formation of Bjerrum pairs in the water layers adjacent to the channel walls. The reduced dielectric constant in these water layers could facilitate this pairing near the walls^{7,10,11}.

Question 3.3

Finally, it is again speculative the existence of "adsorption/desorption" at the surface at the heart of the reported "long-term memory" displayed by all channels. For these reasons, the publication of the manuscript is not recommended.

The adsorption/desorption mechanism as an explanation for crossing memory (M3) was previously **simulated and observed experimentally** in our Science 2023 paper,¹ mainly described for activated carbon nanochannels which are characterized by high surface charge. In the present study, we extend this mechanism to MoS₂ and hBN nanochannels at different experimental conditions (salts, concentration), where **our results continue to strongly correlate with this mechanism**. Additionally, the minimal model that we added in this revision, and the observation of the two crossing memristor behaviours with the variation of the kinetics of adsorption/desorption parameters of the different ions, support our assumption.

In this version, we acknowledged the necessity for direct in-situ experimental verification of adsorption/desorption dynamics at the nanoscale. Spectroscopic techniques adapted to confined nanochannel environments could offer additional insights into these processes. While such techniques are challenging, we believe that our findings provide a solid experimental foundation for future studies in this direction. Additionally, the consistency of our observations across different materials, electrolyte conditions, and confinement regimes provides strong evidence supporting this explanation. Particularly the results of change of memory between Wien and crossing 2 with the variation of pH reflect the role of surface charge in this mechanism.

We added the following phrases to the revised main article page 10:

"In the case of crossing memristors, the kinetic variation was attributed to the adsorption/desorption of ions on the nanochannel walls¹⁰. Previously Robin et al.¹⁰ discussed the case of the adsorption of cations in highly negative charge activated carbon nanochannels and its role in the observation of long-term memory. Here, we observed the crossing 2 phenomena in hBN and MoS₂ nanochannels. However, decreasing the pH of 10 mM KCl solution resulted in the disappearance of crossing 2 (Figure S15), suggesting main role of surface charge and corroborating the suggested mechanism."

Additionally, we have addressed Reviewer 3's concerns regarding the quantitative aspect and simulation of the different memristive phenomena by expanding the discussion in the conclusion in pages 21 to 22.

The four observed memristive phenomena coexist within the same device architecture as demonstrated by the qualitative trends in Figure 2. Given the complex interplay between surface charge effects, ion-ion interactions, and nanochannel confinement and geometry, we developed a minimal model and theory to explain the various observed memristors versus variation of these interaction parameters. The simulation results were corroborated the experimental results. Further, implementation of spectroscopic and in situ characterization methods capable of probing these mechanisms directly at the nanoscale would be of high interest to pursue in future.

We sincerely **hope that the reviewer will reconsider the value of this work**, as it provides key experimental insights and new memory styles that will guide both experimentalists and theorists in further understanding nanofluidic memristors.

References

- 1 Robin, P. *et al.* Long-term memory and synapse-like dynamics in two-dimensional nanofluidic channels. *Science* **379**, 161-167 (2023).
- 2 Kole, M. H. *et al.* Action potential generation requires a high sodium channel density in the axon initial segment. *Nature neuroscience* **11**, 178-186 (2008).
- 3 Bocquet, L. & Charlaix, E. Nanofluidics, from bulk to interfaces. *Chemical Society Reviews* **39**, 1073-1095, doi:10.1039/B909366B (2010).
- 4 Darwent, B. d. (, National Institute of Standards and Technology, Gaithersburg, MD, 1970).
- 5 Conroy, D., Craster, R., Matar, O., Cheng, L.-J. & Chang, H.-C. Nonequilibrium hysteresis and Wien effect water dissociation at a bipolar membrane. *Physical Review E—Statistical, Nonlinear, and Soft Matter Physics* **86**, 056104 (2012).
- 6 Cheng, L.-J. & Chang, H.-C. Switchable pH actuators and 3D integrated salt bridges as new strategies for reconfigurable microfluidic free-flow electrophoretic separation. *Lab on a Chip* **14**, 979-987 (2014).
- 7 Cheng, L.-J. & Chang, H.-C. Microscale pH regulation by splitting water. *Biomicrofluidics* **5** (2011).
- 8 Robin, P. *et al.* Long-term memory and synapse-like dynamics in two-dimensional nanofluidic channels. *Science* **379**, 161-167, doi:doi:10.1126/science.adc9931 (2023).
- 9 Robin, P., Kavokine, N. & Bocquet, L. Modeling of emergent memory and voltage spiking in ionic transport through angstrom-scale slits. *Science* **373**, 687-691 (2021).
- 10 Zhu, H. *et al.* Investigation of dielectric constants of water in a nano-confined pore. *RSC advances* **10**, 8628-8635 (2020).
- 11 Wang, R. *et al.* In-plane dielectric constant and conductivity of confined water. *arXiv preprint arXiv:2407.21538* (2024).

Reply to reviewers

Programmable memristors with two-dimensional nanofluidic channels

Abdulghani Ismail^{1,2}, Gwang-Hyeon Nam^{1,2}, Aziz Lokhandwala,^{1,2} Siddhi Vinayak Pandey,^{1,2} Kalluvadi Veetil Saurav,^{2,3} Yi You,^{1,2} Hiran Jyothilal,^{1,2} Solleti Goutham^{1,2}, Ravalika Sajja^{1,2}, Ashok Keerthi^{2,3,4}, Boya Radha^{1,2,4*}

¹Department of Physics and Astronomy, School of Natural Sciences, The University of Manchester, Manchester M13 9PL, United Kingdom

²National Graphene Institute, The University of Manchester, Manchester M13 9PL, United Kingdom

³Department of Chemistry, School of Natural Sciences, The University of Manchester, Manchester M13 9PL, United Kingdom

⁴Photon Science Institute, The University of Manchester, Manchester M13 9PL, United Kingdom

* Correspondence to be addressed to: radha.boyas@manchester.ac.uk

General comments:

We sincerely thank all the reviewers for their thoughtful, constructive, and encouraging feedback. We deeply appreciate the recognition of the significance and novelty of our work on ***programmable memristors with two-dimensional nanofluidic channels***, as well as the acknowledgment of our additional experiments, theoretical insights, and comprehensive responses. The reviewers' positive recommendations and support are truly motivating, and we are grateful for the opportunity to move this manuscript toward publication. We look forward to contributing further to this exciting field and engaging with the broader scientific community.

Reviewer #1:

The revised manuscript entitled “Programmable memristors with two-dimensional nanofluidic channels” described a switchable memristor with memristive performance of 4 patterns, this switching kinetic discussed herein is of great importance in the field of nanofluidic. Generally, all problems proposed are properly answered and I recommend to publish as it is.

We sincerely thank the reviewer for their positive feedback and kind recommendation. We truly appreciate your recognition of the significance of the switchable memristive performance and the value of the comprehensive responses provided.

Reviewer #2:

The authors have conducted more experiments to demonstrate that their hysteresis is not due to the Wien effect of splitting water, whose hysteretic response has been reported in the papers they now cite. In particular, they show with DI water that they get the usual capacitive loop during voltage scans, without hysteresis. I believe this confirms that their observed hysteresis is due to splitting of Bjerrum pair and not water. It is a new phenomenon and I can now recommend this article for publication in Nature Communication. I believe it will attract many researchers to this field of memristor with transient hysteresis.

We sincerely thank the reviewer for their thoughtful feedback and positive recommendation. We greatly appreciate your recognition of our additional experiments clarifying the origin of the hysteresis.

Reviewer #3:

The authors have added a phenomenological theory that illustrates how the competition between different hypothetical microscopic mechanisms is compatible with the different memristive behaviours. Publication of the manuscript can now be recommended; I am looking forward to seeing in the future direct evidence of these interesting molecular mechanisms.

We are thankful to reviewer for their positive evaluation and recommendation. We appreciate your acknowledgment of the added phenomenological theory explaining the interplay of microscopic mechanisms behind the different memristive behaviours.